# Transcriptome analysis reveals increased abundance and diversity of opportunistic fungal pathogens in nasopharyngeal tract of COVID-19 patients

**M. Nazmul Hoque**[1], **M. Shaminur Rahman**[2], **Md. Murshed Hasan Sarkar**[3], **Md Ahashan Habib**[3], **Shahina Akter**[3], **Tanjina Akhtar Banu**[3], **Barna Goswami**[3], **Iffat Jahan**[3], **M. Anwar Hossain**[4], **M. Salim Khan**[3]*, **Tofazzal Islam**[5]*

1 Department of Gynecology, Obstetrics and Reproductive Health, Bangabandhu Sheikh Mujibur Rahman Agricultural University (BSMRAU), Gazipur, Bangladesh, 2 Department of Microbiology, Jashore University of Science and Technology, Jashore, Bangladesh, 3 Bangladesh Council of Scientific & Industrial Research (BCSIR), Dhanmondi, Dhaka, Bangladesh, 4 Jashore Unive rsity of Science and Technology, Jashore, Bangladesh, 5 Institute of Biotechnology and Genetic Engineering (IBGE), BSMRAU, Gazipur, Bangladesh

☯ These authors contributed equally to this work.
* tofazzalislam@bsmrau.edu.bd (TI); k2salim@yahoo.com (MSK)

**Data Availability Statement:** The sequence data reported in this article has been deposited in the National Center for Biotechnology Information

## Abstract

We previously reported that SARS-CoV-2 infection reduces human nasopharyngeal commensal microbiomes (bacteria, archaea and commensal respiratory viruses) with inclusion of pathobionts. This study aimed to assess the possible changes in the abundance and diversity of resident mycobiome in the nasopharyngeal tract (NT) of humans due to SARS-CoV-2 infections. Twenty-two (n = 22) nasopharyngeal swab samples (including COVID-19 = 8, Recovered = 7, and Healthy = 7) were collected for RNA-sequencing followed by taxonomic profiling of mycobiome. Our analyses indicate that SARS-CoV-2 infection significantly increased (p < 0.05, Wilcoxon test) the population and diversity of fungi in the NT with inclusion of a high proportion of opportunistic pathogens. We detected 863 fungal species including 533, 445, and 188 species in COVID-19, Recovered, and Healthy individuals, respectively that indicate a distinct mycobiome dysbiosis due to the SARS-CoV-2 infection. Remarkably, 37% of the fungal species were exclusively associated with SARS-CoV-2 infection, where *S. cerevisiae* (88.62%) and *Phaffia rhodozyma* (10.30%) were two top abundant species. Likewise, Recovered humans NT samples were predominated by *Aspergillus penicillioides* (36.64%), *A. keveii* (23.36%), *A. oryzae* (10.05%) and *A. pseudoglaucus* (4.42%). Conversely, *Nannochloropsis oceanica* (47.93%), *Saccharomyces pastorianus* (34.42%), and *S. cerevisiae* (2.80%) were the top abundant fungal species in Healthy controls nasal swabs. Importantly, 16% commensal fungal species found in the Healthy controls were not detected in either COVID-19 patients or when they were cured from COVID-19 (Recovered). We also detected several altered metabolic pathways correlated with the dysbiosis of fungal mycobiota in COVID-19 patients. Our results suggest that SARS-CoV-2 infection causes significant dysbiosis of mycobiome and related metabolic functions possibly play a determining role in the progression of SARS-CoV-2 pathogenesis. These findings

(NCBI) under BioProject accession number PRJNA720904.

**Funding:** This project is financed by the Ministry of Science and Technology, Government of the People's Republic of Bangladesh.

**Competing interests:** The authors have declared that no competing interests exist.

might be helpful for developing mycobiome-based diagnostics, and also devising appropriate therapeutic regimens including antifungal drugs for prevention and control of concurrent fungal coinfections in COVID-19 patients.

## Introduction

Coronavirus disease (COVID-19), emerged as one of the deadliest human diseases, is considered as the fast expanding pandemics since the 1918 Spanish flu with serious consequences for global health and economy [1–3]. Since SARS-CoV-2 emerged in the human population, the global scientific community is working round the clock to find good strategies for the containment and treatment of this pandemic virus, SARS-CoV-2 [4]. Upon inhalation, SARS-CoV-2 primarily enters the nasal epithelial cells of the human NT through the ACE2 and TMPRSS2 receptors [5], and then gradually move towards the lung to initiate infection followed by onset of acute respiratory distress syndrome (ARDS) [6]. The capability of the SARS-CoV-2 for swiftly adapting to diverse environments of the host body could be linked with coinfecting pathogens [7, 8]. Viral replication in the nasopharyngeal epithelial cells elicits direct adverse effects on resilient microbiomes [4, 9], and induces local immune cells to quickly and abundantly secrete cytokines and chemokines [10]. Subsequently, severe lung damage and immunologic derangement resulting from SARS-CoV-2 infection or its treatment predispose to coinfections with multiple pathogens, including bacteria, other viruses and fungi [11–13]. Clinical trials and high throughput sequencing (metagenomic and RNA-seq)-based investigations on SARS-CoV-2 revealed that severely and non-severely ill COVID-19 patients had coinfections with respiratory viral pathogens [14], and bacteria and/or fungi [11, 13, 15, 16]. A retrospective study found that the coinfection rate of SARS-CoV-2 and influenza virus was as high as 57.3% in COVID-19 patients during the outbreak period in Wuhan [17]. The coinfection in COVID-19 patients may be a predisposing factor of increased morbidity and mortality rates throughout the globe [11, 14, 18]. Previously, the SARS outbreak was characterised by an high rate of nosocomial transmission of drug-resistant microorganisms [13, 19].

Fungal infections are known to be among the infectious complications related to the damage caused by viral pulmonary infections, particularly in patients admitted to intensive care units with ARDS [11]. Patients with severe COVID-19 have also emerged as a population with a high risk of fungal infections [11, 20]. There are reports that immunocompromised COVID-19 patients were at a higher risk of development of mysterious fungal infection known as mucormycosis or black fungus [21, 22]. Recently other non-Aspergillus fungal coinfections, including mucormycosis in India, have been reported in those with severe COVID-19 pulmonary disease [23]. Meanwhile, a descriptive study held by Chen et al. (2020) showed that the coinfected fungi includes *Aspergillus* spp., *Candida albicans*, and *Candida glabrata* [24]. Fungal coinfection was the main cause of death for SARS patients, accounting for 25–73.7% in all causes of death [25]. Besides, in the past decade, increasing reports of severe influenza pneumonia resulting in ARDS complicated by fungal infection were published [26]. With the aggravating pathogenesis of SARS-CoV-2, most of the COVID-19 patients usually undergone to the in-time use of broad spectrum antibiotics, dexamethasone, and immunosuppressive therapies with corticosteroids or immunomodulators for bacterial coinfections [20, 27], while the diagnosis of fungal coinfection is always delayed or neglected. Based on the experience of SARS in 2003 and the cases of invasive aspergillosis combined with severe influenza, it is critically important to pay attention to the probability of COVID-19 accompanied by fungal infections.

However, as for fungal coinfection in COVID-19 patients, only few studies have reported it, which may have been neglected. Clinically, many COVID-19 patients did not undergo fungal assessment at the beginning, moreover, it is difficult to detect fungus with a single sputum fungal culture [25].

Human microbiota plays a critical role in immunity and health of individual hosts, and thus, microbiome dysbiosis in the respiratory tract by the pathogenic virus like SARS-CoV-2 can increase the mortality rate in patients [13, 28, 29]. Coinfection can also change the upper airway microbiome homeostasis and thus, triggers the infection and stimulates immune cells to produce more severe inflammation [11, 13, 30]. Therefore, we hypothesize that during this migration, propagation and immune response, the inhabitant fungal microbiomes in the respiratory airways are altered, and inclusion of some of the fungal pathobionts might aggravate the progression and lethality in COVID-19 patients. Therefore, timely diagnosis of coinfecting fungal pathogen(s) in COVID-19 patients is important to initiate appropriate therapy and limit the overuse of antimicrobial agents. To shed light on the effects and consequences of SARS-CoV-2 infections on changes in the resident mycobiome in the NT, we conducted a high throughput RNA-seq analysis of the nasopharyngeal swabs randomly collected from Healthy controls, COVID-19 patients and COVID-19 Recovered individuals. Our results demonstrate that SARS-CoV-2 infection is critical for inclusion of opportunistic mycobiota with loss of salutary fungi in the NT. Besides, we conducted a comparative metabolic functional analysis to identify the potential biological mechanisms linking the shift of fungal population in NT, and their correlation with differentially abundant fungal taxa in the COVID-19 patients, Recovered humans and Healthy controls. Taken together, our data suggest a critical association of altered mycobiome in the pathophysiology of SARS-CoV-2 infections in the nasal cavity of COVID-19 patients.

## Materials and methods

### Subject recruitment and sample collection

We collected twenty-two (n = 22) nasopharyngeal samples (including COVID-19 = 8, Recovered = 7, and Healthy = 7) from Dhaka city of Bangladesh during May to July, 2020. The suspected patients were diagnosed positive for SARS-CoV-2 infections (COVID-19) through RT-qPCR. The confirmed patients were admitted into the dedicated COVID-19 isolation wards, and received medication. These patients were tested negative for COVID-19 after 17.5 (ranged from11 to 32) days of SARS-CoV-2 infection, and categorized as Recovered humans (S1 Table). The nasopharyngeal swab samples from COVID-19 and Recovered subjects were collected on the test day (COVID-19 positive and COVID-19 negative confirmation day). The Healthy control subjects were randomly selected and these people did not show any signs and symptoms of respiratory illness. Nasopharyngeal swabs from these Healthy people were collected following the same protocol for COVID-19 and Recovered humans. The collected samples were placed in sample collection vial containing PBS (phosphate buffered saline). The RT-qPCR was performed for *ORF1ab* and *N* genes of SARS-CoV-2 using novel Coronavirus (2019-nCoV) Nucleic Acid Diagnostic Kit (PCR-Fluorescence Probing, Sansure Biotech Inc.) following the manufacturer's instructions. The collected samples were immediately sent for RNA extraction, library preparation and sequencing [13].

### RNA extraction and sequencing

Total RNA was extracted using a PureLink viral RNA/DNA minikit (Thermo Fisher Scientific, USA). RNA was extracted from a 20 μL swab sample through lysis with sample release reagent provided by the kit and then directly used for RT-qPCR. A thermal cycling of 50°C for 30 min

was performed for reverse transcription, followed by 95˚C for 1 min, and then 45 cycles of 95˚C for 15 s, 60˚C for 30 s on an Analytik-Jena qTOWER instrument (Analytik Jena, Germany). RNA-seq libraries were prepared from total RNA using the Nextera DNA Flex library preparation kit (Illumina, Inc., San Diego, CA) according to the manufacturer's instructions where the first-strand cDNA was synthesized using SuperScript II Reverse Transcriptase (Thermo Fisher) and random primers. Paired-end (2 × 150 bp reads) sequencing of the prepared RNA library pools was performed using a NextSeq high throughput kit under Illumina platform with an Illumina NextSeq 550 sequencer at the Genomic Research Laboratory, Bangladesh Council of Scientific and Industrial Research (BCSIR), Dhaka, Bangladesh [13].

## Data processing and taxonomic identification of fungal communities

The raw sequencing reads generated from Illumina platform were adapter and quality trimmed through BBDuk (with options k = 21, mink = 6, ktrim = r, ftm = 5, qtrim = rl, trimq = 20, minlen = 30, overwrite = true) [31]. Any sequence below these thresholds or reads containing more than one 'N' were discarded [31]. The good quality reads from COVID-19, Recovered and Healthy samples (n = 22) were analyzed using two different bioinformatics tools: the IDSeq (an open-source cloud-based pipeline to assign taxonomy) [32] and the MG-RAST (release version 4.1) (MR) and both use mapping and assembly-based hybrid method [33]. IDseq- an open-source cloud-based pipeline has been used to assign taxonomy with NT L (nucleotide alignment length in bp) > = 50 and NT %id > = 97 [13]. This pipeline used quality control, host filtering, assembly-based alignment and taxonomic reporting aligning to NCBI nucleotide database. In MR analysis, the uploaded reads were subjected to optional quality filtering with dereplication and host DNA removal, and finally functional assignment. For this pipeline, we employed the "Best Hit Classification" option to determine taxonomic abundance using the NCBI database as a reference with the following set parameters: maximum $e$-value of $1 \times 10^{-30}$; minimum identity of 90% using a minimum alignment length of 20 as the set parameters. Microbial taxa that were detected in one group of sample but not detected in rest of the two groups are denoted as solely (unique) associated microbiomes [31]. We simultaneously uploaded the filtered sequence data in both pipelines with proper embedded metadata.

## Functional profiling of the nasopharyngeal mycobiome

We performed the fungal metabolic functional classification through mapping the reads onto the Kyoto Encyclopaedia of Genes and Genomes (KEGG) database [34], and SEED subsystem identifiers [33] of the MR server using the partially modified set parameters ($e$-value cut off: $1 \times 10^{-30}$, min. % identity cut off: 90%, and min. alignment length cut off: 20) [35]. The association between metabolic functions and dominant fungal species were measured using the Spearman's correlation coefficient and significance tests [36]. The R packages, Hmisc (https://cran.r-project.org/web/packages/Hmisc/index.html) and corrplot (https://cran.r-project.org/web/packages/corrplot/vignettes/corrplot-intro.html) were used respectively to analyse and visualize the data.

## Statistical analysis

Read normalization in each sample was performed using median sequencing depth through Phyloseq (version 4.1) package in R [37]. We calculated the alpha diversity (diversity within samples) using the observed species, Chao1, ACE, Shannon, Simpson and InvSimpson diversity indices, and performed the non-parametric Wilcoxon rank-sum test to evaluate diversity differences in different samples. A principal coordinate analysis (PCoA) based on the Bray-

Curtis distance method was performed to visualize differences in fungal diversity across three metagenomes. To calculate the significance of variability patterns of the mycobiome (generated between sample categories), we performed PERMANOVA (permutational multivariate analysis of variance) using 999 permutations on all three sample types at the same time and compared them pairwise. For these statistical analyses, pairwise non-parametric Wilcoxon test was performed using the Phyloseq and Vegan (package 2.5.1 of R 3.4.2) programs [38]. Dominant fungal community were determined with $\geq$ 1% median relative abundance by groups. After filtering, 11 taxa remained for which Spearman's correlation analysis between KEGG pathways and SEED functions pathways was done in Hmisc's rcorr function [39] and the corrplot function [40] of the corrplot R package as mentioned in the previous section. In addition, Kruskal-Wallis test was also applied at different KEGG and SEED subsystems levels through IBM SPSS (SPSS, Version 23.0, IBM Corp., NY, USA) [31].

## Results

In order to detect the dysbiosis of the inhabitant mycobiome in the NT after SARS-CoV-2 infection, we analysed 22 nasal swab samples from Healthy individuals, COVID-19 patients and Recovered humans. The demographics, health-related characteristics, and symptomatology of the study subjects are described in S1 Table. Overall, 15 people were male (68.18%) and seven were female (31.82%) with a median age of 40.87 years. In this study, patients were diagnosed positive for SARS-CoV-2 infections (COVID-19) on an average 5.6 days after the onset of pneumonia like clinical symptoms, and these patients tested negative for SARS-CoV-2 (Recovered) on an average 15.12 days after the initial COVID-19 confirmatory diagnosis. The confirmed COVID-19 patients received medication for an average period of 15.71 days. The Healthy control subjects however did not show any signs and symptoms of respiratory illness (S1 Table). We were able to characterize both common and differentially abundant fungal taxa in each sample groups along with concurrent metabolic functional perturbations.

### SARS-CoV-2 infection alters mycobiome diversity in the nasopharyngeal tract

To understand whether SARS-CoV-2 infection alters the composition and diversity of the NT mycobiome, we examined both within sample (alpha) and across the samples (beta) diversities of the detected fungal community in Healthy human, COVID-19 patients and Recovered humans (Fig 1). The alpha diversity measured using Observed, Chao1, ACE, Shannon, Simpson and InvSimpson indices showed significant differences in fungal community richness, keeping substantially higher diversity in the microbial niche of COVID-19 (p = 0.01; Wilcoxon test) followed by Recovered (p = 0.05; Wilcoxon test) and Healthy (p > 0.05; Wilcoxon test) samples (Fig 1A). The Bray–Curtis dissimilarity distance estimated principal coordinate analysis (PCoA) plot showed that fungal composition in the NT nasal cavity differed according to sample categories of the study people (Fig 1B). The beta diversity of microbiomes however did not vary significantly (p > 0.05, PERMANOVA test) according to the sex (male or female) of the study people (Fig 1B).

The unique and shared distribution of fungal taxa found in the three metagenome groups is presented by comprehensive Venn diagrams (Fig 2). A total of 28 fungal phyla were detected in three metagenomes including 16, 24 and 18 phyla in Healthy, COVID-19 and Recovered samples, respectively. Among these phyla, six phyla had sole association with SARS-CoV-2 infections, and 10 were found to be shared across three sample groups (Fig 2A, S1 Data). We detected 190 orders of fungi including 78, 131 and 104 in Healthy, COVID-19, and Recovered nasopharyngeal samples, respectively, and of them, 52 orders had sole association with

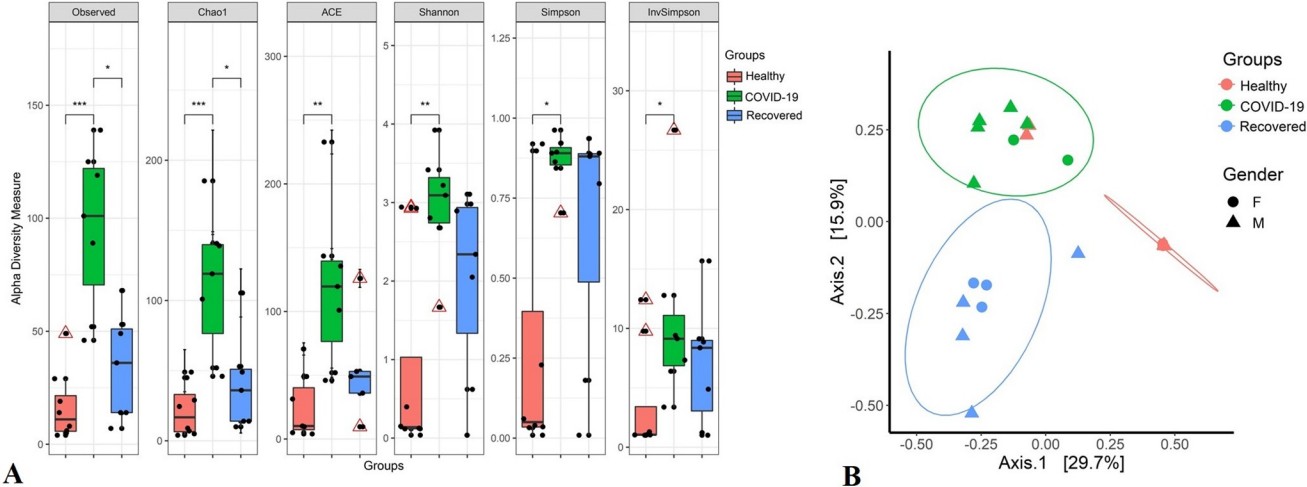

**Fig 1. Mycobiome diversity.** (A) Within subject (Alpha) diversity measure. Observed species, Chao1, ACE, Shannon, Simpson and InvSimpson indices estimated within sample fungal diversity of Healthy, COVID-19 and Recovered cases are plotted on boxplots and comparisons are made with pairwise Wilcoxon rank sum tests. Significance level (p-value) 0.0001, 0.001, 0.01, and 0.05 are represented by the symbols "****", "***", "**", and "*", respectively. (B) Between subject (Beta) diversity measure. Fungal beta diversity is measured with Bray-Curtis dissimilarity distances and visualized on principal coordinate analysis (PCoA) plot. The samples are colored according to subject groups (e.g., red: Healthy, green: COVID-19 and blue: Recovered) and joined with the respective ellipses. The shapes represent the gender of the assigned subjects: circular for female (F) and triangular for male (M). Pairwise comparisons on a distance matrix using PERMANOVA test under reduced model shows significant fungal community differences among the groups (p < 0.01, PERMANOVA test).

SARS-CoV-2 infection, and only 35 were common in three sample groups (Fig 2B, S1 Data). Likewise, 532 fungal genera were identified, of which 57, 213 and 128 genera had sole association with Healthy, COVID-19, and Recovered subjects, respectively, and only 34 genera were found to be shared across three metagenomes (Fig 2C, S1 Data). One of the noteworthy findings of the present study was the detection of 862 fungal species and of them, 188, 533 and 445 species were found in Healthy, COVID-19, and Recovered samples, respectively. However, among the detected fungal species, only 65 (7.54%) were found to be shared across the given conditions (Fig 2D). Remarkably, compared to Healthy controls and Recovered cases, COVID-19 patients had sole association of higher number of fungal species (n = 315, 36.54%) which probably due to the opportunistic inclusion during the pathogenesis of SARS-CoV-2 infection. Similarly, Recovered humans swab samples had sole association of 227 (26.33%) fungal species revealing the re-establishment beneficial commensal flora after the recovery of SARS-CoV-2 infections. Conversely, 81 fungal species had sole association with healthy states (Healthy control) of the humans (Fig 2D), and none of these species was detected in the COVID-19 patients swab samples demonstrating that these commensal microbes underwent to dysbiosis through the effect of SARS-CoV-2 infection (Fig 2D, S1 Data).

## SARS-CoV-2 infection induces nasopharyngeal mycobiome dysbiosis

To determine whether SARS-CoV-2 infection induces dysbiosis of the NT mycobiome, we characterized fungal taxa at different taxonomic ranks (phylum to species level) across three metagenomes. In this study, fungal communities in COVID-19 patients, Recovered humans and Healthy control samples were predominated by *Ascomycota* (> 87.0%) phylum, however, other abundant phyla were *Basidiomycota* (2.26 to 3.83%), *Streptophyta* (1.41 to 2.20%), and *Mucoromycota* (2.06%) (S1 Fig). Rest of the fungal phyla detected in all of the metagenomes had relatively lower abundances (< 1.0%) (S1 Data). Moreover, the average phyla distribution

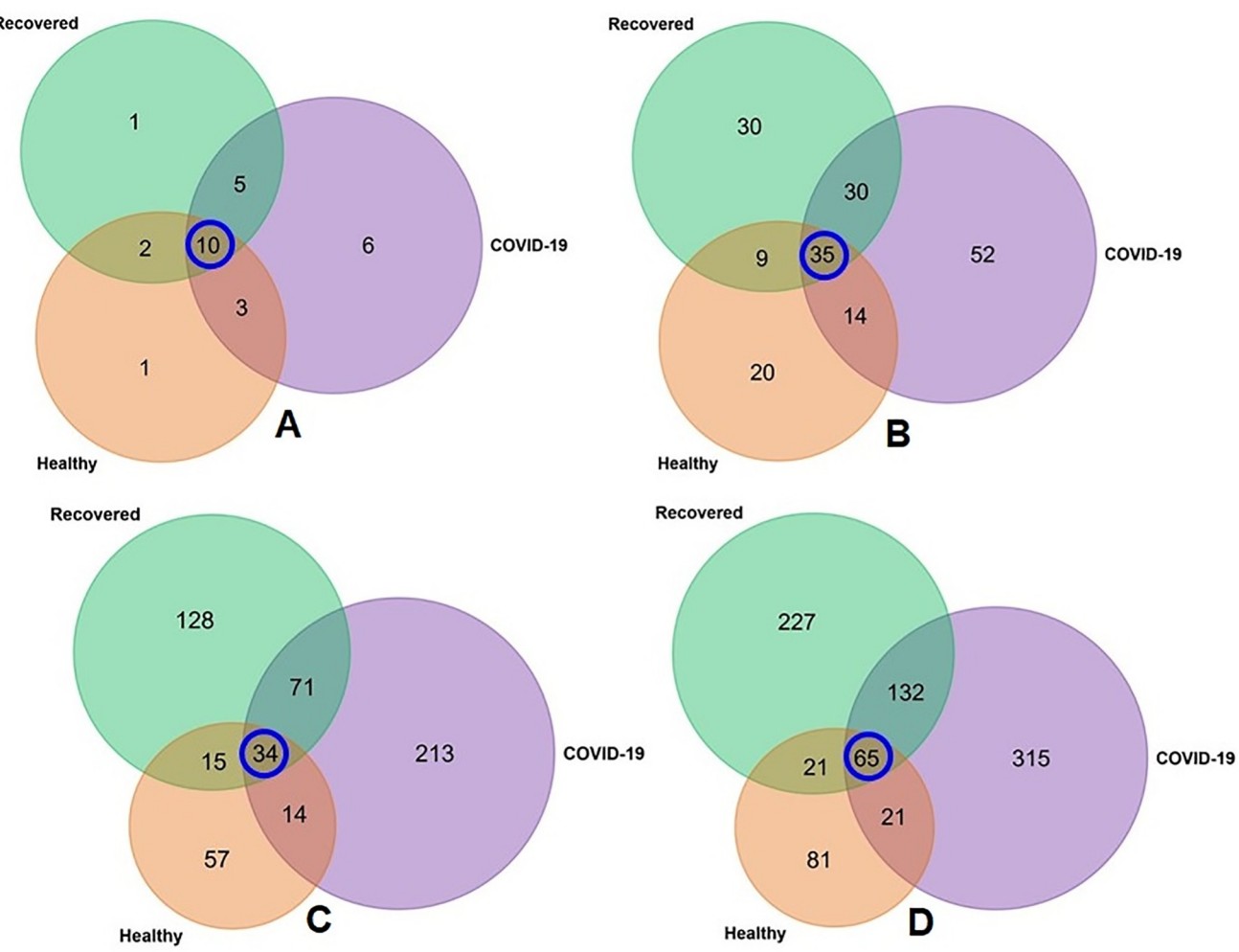

**Fig 2. Taxonomic composition of mycobiome.** Venn diagrams representing the unique and shared fungal taxa in Healthy, COVID-19, and Recovered nasopharyngeal sample groups. (A) Venn diagram showing unique and shared fungal phyla. Out of 28 detected phyla, only 10 phyla (highlighted in blue circle) were found to be shared in the metagenomes. (B) Venn diagram comparison of 190 orders of fungi detected across the sample groups, of which only 35 (highlighted in blue circle) orders were found to be shared among the conditions. (C) Venn diagrams representing unique and shared fungal genera identified in three metagenomes. Of the detected fungal genera (n = 532), 57, 213 and 128 genera had sole association with Healthy, COVID-19, and Recovered subjects, respectively, and only 34 genera (highlighted in blue circle) were found to be shared across three metagenomes. (D) Venn diagrams representing unique and shared fungal species identified in three metagenomes. Of the detected fungal species (n = 862), the Healthy, COVID-19 and Recovered cases had sole association of 81, 227 and 315 species, respectively, and 65 species (highlighted in blue circle) were found to be shared across the study sample groups. More information on the taxonomic result is also available in S1 Data.

in SARS-CoV-2 infection associated metagenomes (COVID-19 and Recovered) was different compared to that in Healthy controls. Notably, the distribution phyla in COVID-19 and Recovered cases demonstrated greater similarities than those detected in Healthy controls (S1 Fig).

We then pairwise compared the NT fungal phyla between the subjects with SARS-CoV-2 infection (COVID-19) and without SARS-CoV-2 infection (Healthy and Recovered) on a distance matrix using PERMANOVA test under reduced model which showed significant differences ($p < 0.01$, PERMANOVA test) in microbial community across the study groups. Pairwise Wilcoxon tests identified that five phyla (*Ascomycota*, *Streptophyta*, *Tubulinea*, *Bacillariophyta* and *Evosea*) were significantly different ($p < 0.05$, Wilcoxon test) in the Healthy,

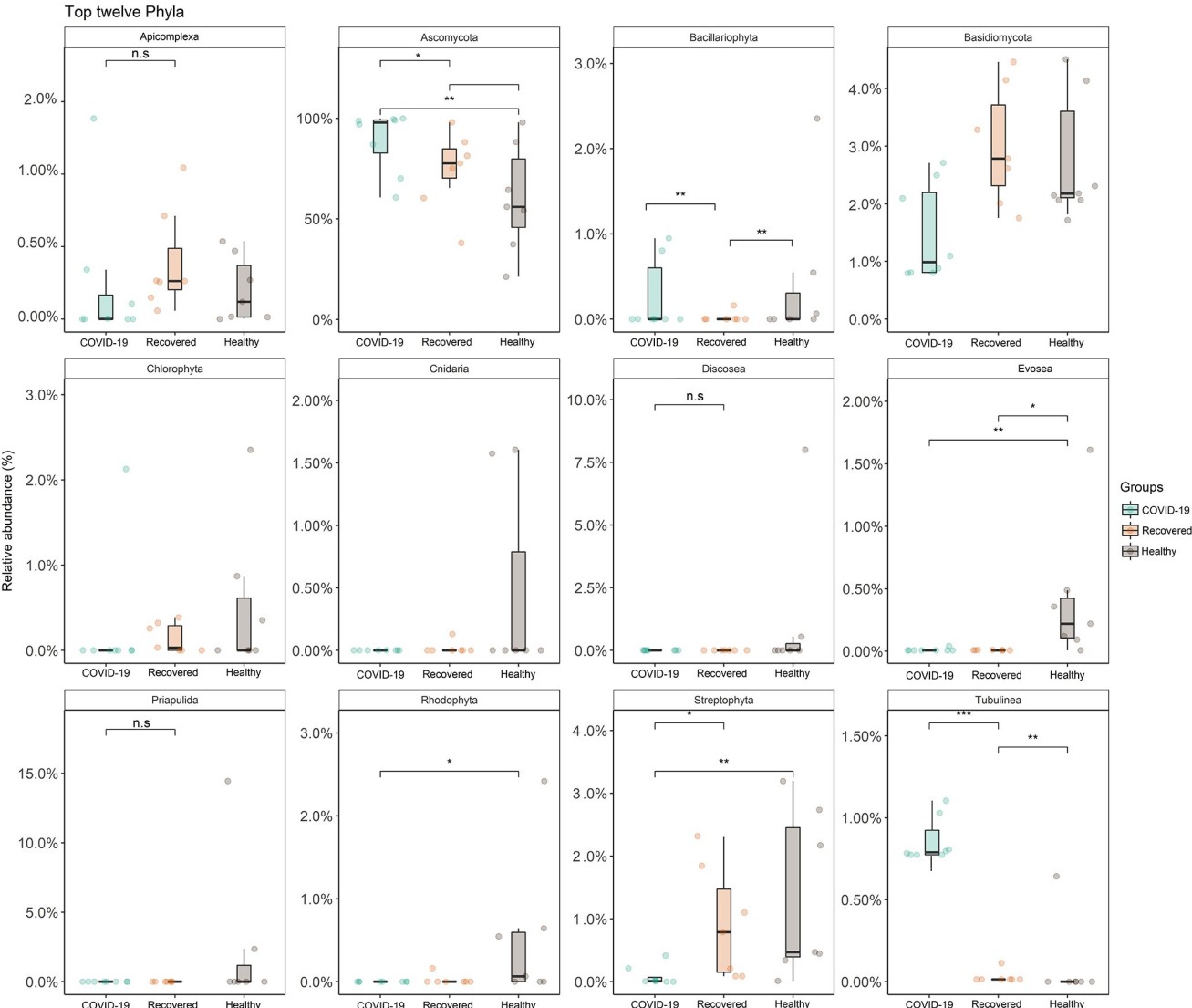

**Fig 3. Top twelve fungal phyla detected.** The phylum-level taxonomic abundance of fungal microbiomes in COVID-19, Recovered and Healthy nasopharyngeal samples. The diversity for each phylum is plotted on boxplots and comparisons are made with pairwise Wilcoxon test rank sum tests. Significance level (p-value) 0.0001, 0.001, 0.01, 0.05, and 0.1 are represented by the symbols "****", "***", "**", "*", and "n.s", respectively.

COVID-19 and Recovered metagenomes (Fig 3). Healthy and Recovered samples had significantly higher (p = 0.01, Wilcoxon test) relative abundance of *Streptophyta* than COVID-19 samples (p = 0.05, Wilcoxon test). In addition to *Ascomycota* (96.70%), the COVID-19 samples had significantly higher (p < 0.05, Wilcoxon test) relative abundance of *Tubulinea* and *Bacillariophyta* (Fig 3). The Recovered metagenome had significantly higher relative abundance of *Mucoromycota* (2.06%) and *Apicomplexa* (0.32%) compared to Healthy controls and COVID-19 patients (≤ 0.01% in both). Conversely, the Healthy humans NT swab samples had higher relative abundances of *Cnidaria*, *Chlorophyta*, *Discosea*, *Evosea* and *Rhodophyta* compared to COVID-19 and Recovered samples (Fig 3, S1 Data).

We also demonstrated notable differences in both composition and the relative abundances of fungal taxa at genus-level among COVID-19 patients, Recovered humans and Healthy

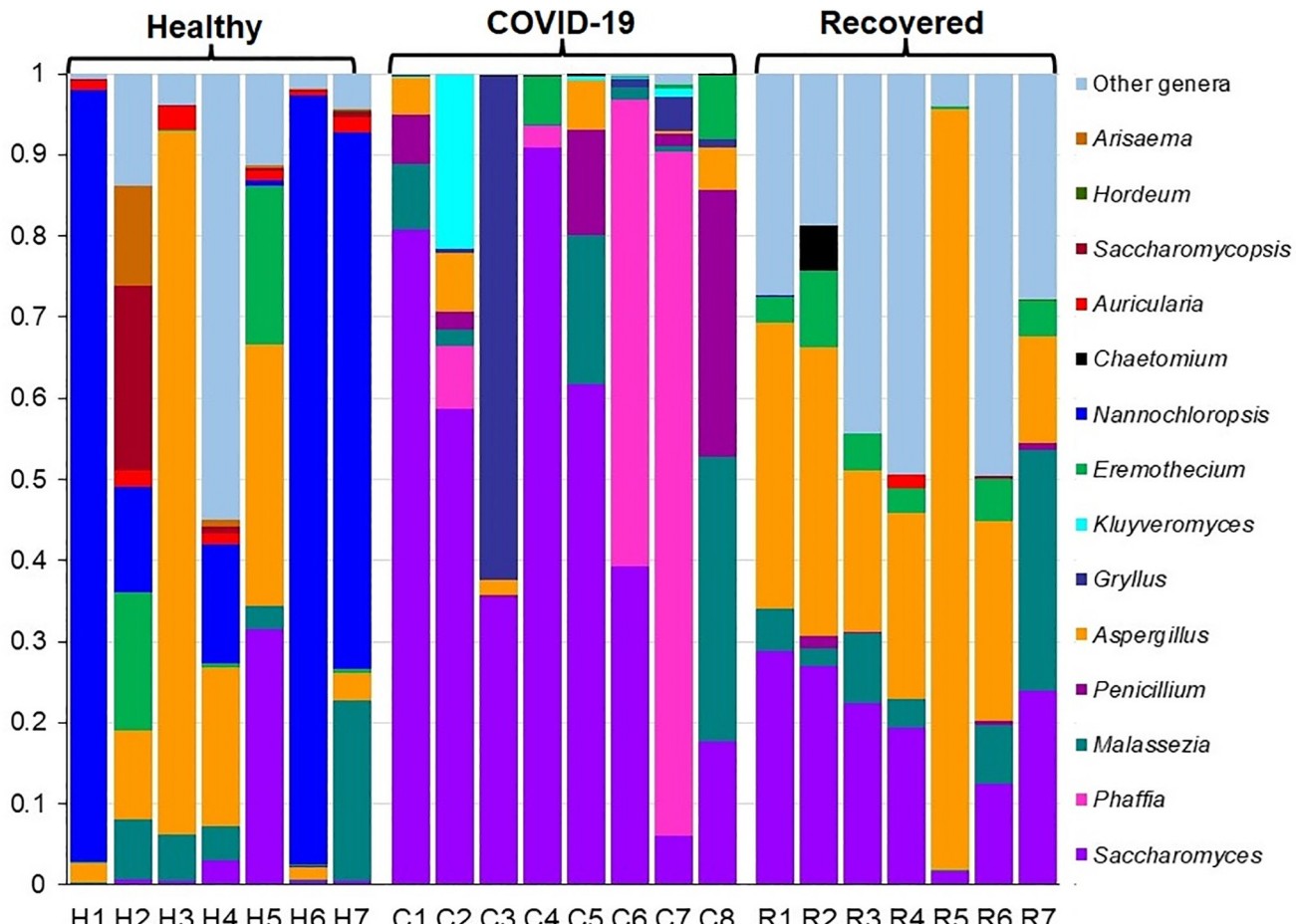

**Fig 4. The genus-level taxonomic profile of mycobiome.** The bar plots representing the relative abundance of 15 top abundant fungal genera in Healthy (H1-H7), COVID-19 (C1-C8), and Recovered (R1-R7) human nasopharyngeal samples. Fourteen genera are sorted from bottom to top by their deceasing proportion of the mean relative abundances, with the remaining genera keeping as 'Other genera'. Each stacked bar plot represents the abundance of fungal genera in each sample of the corresponding category. Notable differences in fungal populations are those where the taxon is abundant in COVID-19 and Recovered samples, and effectively not detected in the Healthy controls. The distribution and relative abundance of the fungal genera in the study metagenomes are also available in S1 Data.

controls. The relative abundances of the top 15 fungal genera were compared among the Healthy, COVID-19 and Recovered cohorts (Fig 4). Among these predominating genera, *Nannochloropsis* (81.58%) was the top abundant genus in Healthy controls while *Saccharomyces* (96.49%) and *Aspergillus* (81.90%) were the predominating genera in COVID-19 patients and Recovered humans NT swabs, respectively (Fig 4, S1 Data). The other predominant fungal genera in Healthy controls were *Aspergillus* (3.19%), *Saccharomyces* (2.60%), *Triticum* (2.17%), *Auricularia* (1.30%) and *Eremothecium* (1.10%). Conversely, *Phaffia* (2.88%) in COVID-19 patients, and *Saccharomyces* (4.71%), *Malassezia* (3.49%), *Triticum* (1.69%), and *Eremothecium* (1.20%) in Recovered humans were other abundant fungal genera. Though rest of the genera had relatively lower abundances (<1.0%), but their relative abundances differed across three sample groups (Fig 4, S1 Data). Fungal genera identified in the Healthy humans NT swabs resemble more similarity to those detected in Recovered humans NT swab when compared with those detected in COVID-19 patients NT swabs (S1 Data).

## Differentially abundant and altered fungal species are correlated with COVID-19 pathophysiology

To examine whether species level composition and relative abundance of the fungal taxa statistically differ across the sample groups, we examined pairwise Spearman correlation of abundance of all taxa identified. This differential analysis revealed that genus level mycobiome composition and diversity discrepancy was more evident at species level. However, presence of few predominating fungal species in each sample category suggested that the crucial differences might be found at the strain level. The Healthy controls nasal swab samples were dominated by *Nannochloropsis oceanica* (47.93%), *Saccharomyces pastorianus* (34.42%), *Saccharomyces cerevisiae* (2.80%), *Aspergillus pseudoglaucus* (1.84%), *Aspergillus penicillioides* (1.25%), *Paecilomyces variotii* (1.24%), and *Eremothecium gossypii* (1.06%) (Fig 5, S1 Data). Despite, 36.54% of the fungal species were exclusively associated with SARS-CoV-2 infection, only *S. cerevisiae* (88.62%) was detected as the most predominating species in COVID-19 patients. However, other abundant species identified in this metagenome included *Phaffia*

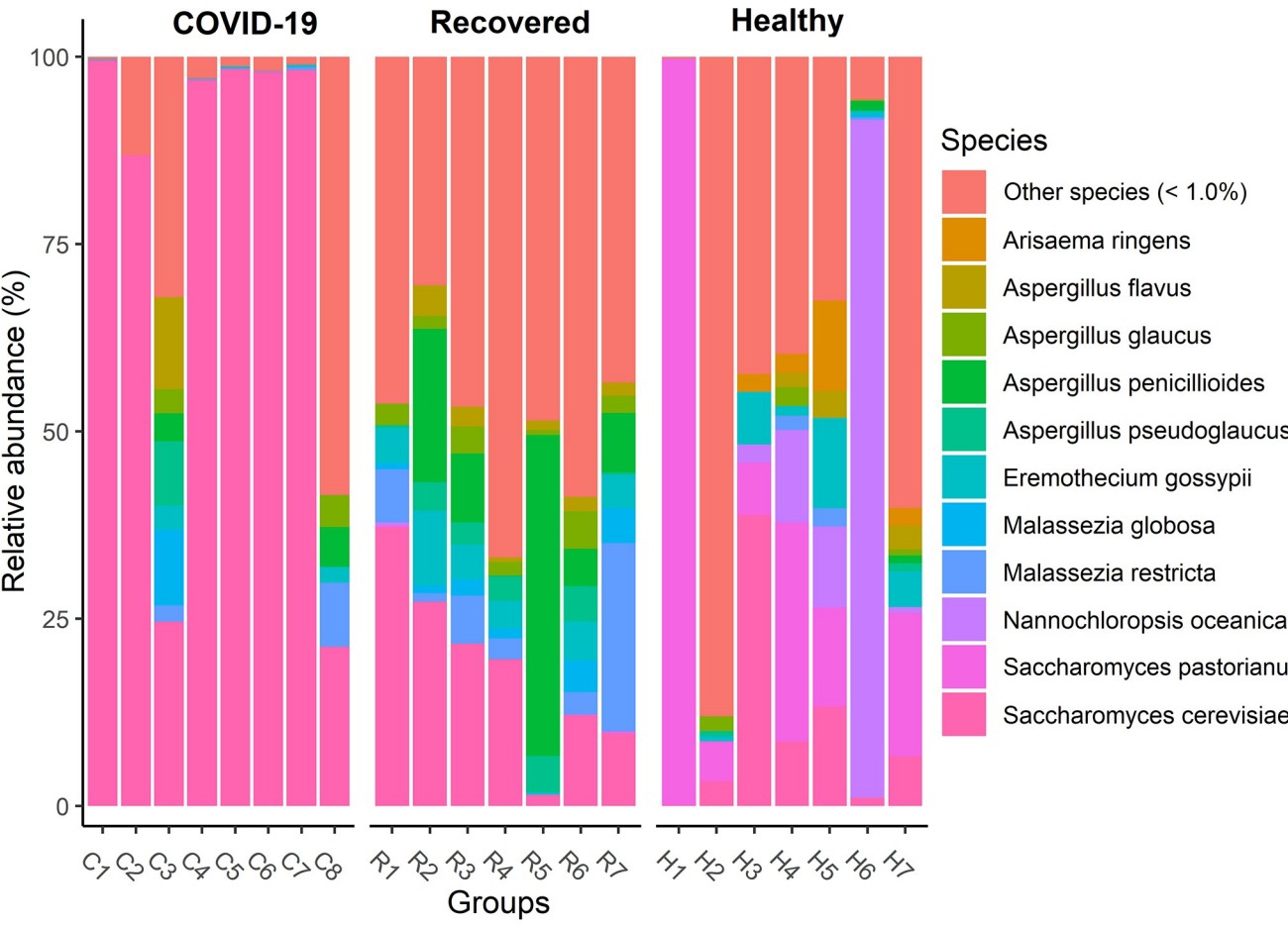

**Fig 5. Major fungal species detected.** The species-level taxonomic profile of mycobiome in COVID-19 (C1-C8), Recovered (R1-R7) and Healthy (H1-H7) nasopharyngeal samples. Species with > 1% mean relative abundance are represented by different color codes against respective sample groups. Other species (< 1%) indicate the rare less abundant taxa in each group, with a mean relative abundance of < 1%. Each stacked bar plot represents the abundance of fungal species in each sample of the corresponding category. Notable differences in fungal populations are those where the species is abundant in COVID-19 and Recovered samples, and effectively not detected in the Healthy controls. The distribution and relative abundance of the fungal genera in the study metagenomes are also available in S1 Data.

*rhodozyma* (10.30%), *S. pastorianus* (0.43%), *Paecilomyces variotii* (0.37%), and *A. pseudoglaucus* (0.17%) (Fig 5, S1 Data). Rest of the species had relatively lower abundances (< 0.1%) in COVID-19 metagenome and possibly played an opportunistic role in the SARS-CoV-2 pathogenesis (S1 Data). In addition, Recovered humans NT swab samples were mostly dominated by different species of *Aspergillus* genus (> 80.0%) such as *A. penicillioides* (36.64%), *A. keveii* (23.36%), *A. oryzae* (10.05%), *A. pseudoglaucus* (4.42%), *A. flavus* (1.44%), *A. fumigatus* (1.34%), *A. glaucus* (1.16%) and *A. lentulus* (1.10%). However, other dominating fungal species in this metagenome were *S. cerevisiae* (4.66%), *Malassezia restricta* (2.63%), *E. gossypii* (1.20%), *P. variotii* (1.08%), and rest of the genera had relatively lower abundances (< 1.0%) (Fig 5, S1 Data).

Our primary microbiome compositional analysis identified 11 species as differentially abundant across Healthy, COVID-19 and Recovered metagenomes. The pairwise statistical relationship analysis of the relative abundances of these 11 taxa and health biomarkers (Healthy, SARS-CoV-2 infection and SARS-CoV-2 recovery) of the study participants showed significant variations (p ≤ 0.05, Wilcoxon test) in 11 differentially abundant fungal species (Fig 6). For instance, *N. oceanica* (p = 0.01, Wilcoxon test) and *S. pastorianus* (p = 0.05, Wilcoxon test) were the two significant and differentially abundant fungal species in Healthy controls compared to either COVID-19 patients or Recovered humans (Fig 6). The COVID-19 samples however harboured only one significantly abundant fungal species, *S. cerevisiae* (p = 0.01, Wilcoxon test) over the Healthy controls or Recovered humans. The Recovered humans however had significantly higher relative abundances of *A. penicillioides* (p = 0.01), *A. pseudoglaucus* (p = 0.01), *A. flavus* (p = 0.01), *A. fumigatus* (p = 0.05) and *E. gossypii* (p = 0.05) compared to either COVID-19 patients or Healthy controls (Fig 6). Similarly, for those species that were less abundant in three metagenomes, we also observed a stronger correlation among the microbiomes of COVID-19 patients and Recovered humans (Fig 6, S1 Data).

## Phylogenetic relatedness of the mycobiome of the nasopharyngeal tract

The unrooted maximum likelihood tree exhibited a species-level (n = 50 top abundant species) topology that was completely congruent with SARS-CoV-2 infection mediated mycobiome dysbiosis (Fig 7). The resulting species tree provided high resolution of the basal relationships among fungal clades, and enabled us to evaluate the established taxonomic hierarchies. Phylogenetic analysis showed that out of 50 species, 13 descended from *Eurotiales* (26.0%) followed by 10 from *Saccharomycetales* (20.0%), four from *Auriculariales* (8.0%), one from *Eustigmatales* (2.0%) and rest of the species fall into other orders (44.0%) (Fig 7, S2 Table). Despite stronger associations among these 50 species used for phylogenetic tree reconstruction, SARS-CoV-2 infection facilitated the opportunistic inclusion of 36.54% fungal species including *S. paradoxus*, *S. kudriavzevii*, *K. lactis*, *K. aquatica*, *Y. lipolytica*, *C. gloeosporioides*, *P. album*, *C. sphaerospermum*, *F. flavus*, *P. hubeiensis*, *P. rhodozyma*, *Tarsonemidae sp. AD1063*, *C. remanei*, *T. govanianum*, and *G. rogosa* (Fig 7, S2 Table). Among these species, *P. rhodozyma* was found as the top abundant (10.30%) opportunistic pathogen in COVID-19 samples (S1 Data). None of these species was detected in the Healthy control samples. Simultaneously, SARS-CoV-2 inflammation was also associated with depletion of 16.0% commensal fungal population such as *E. tenuicula*, *P. caudatus*, *A. ringens*, *N. nucifera*, *A. castellanii*, *M. brevicollis*, *V. Vermiformis*, *A.polytricha* etc. (Fig 7, S2 Table). These species were solely present in Healthy humans nasal cavity, and not merely detected in COVID-19 patients samples Moreover, changes in mycobiome composition were supported by high bootstrap values (98%–100% for each species) (Fig 7).

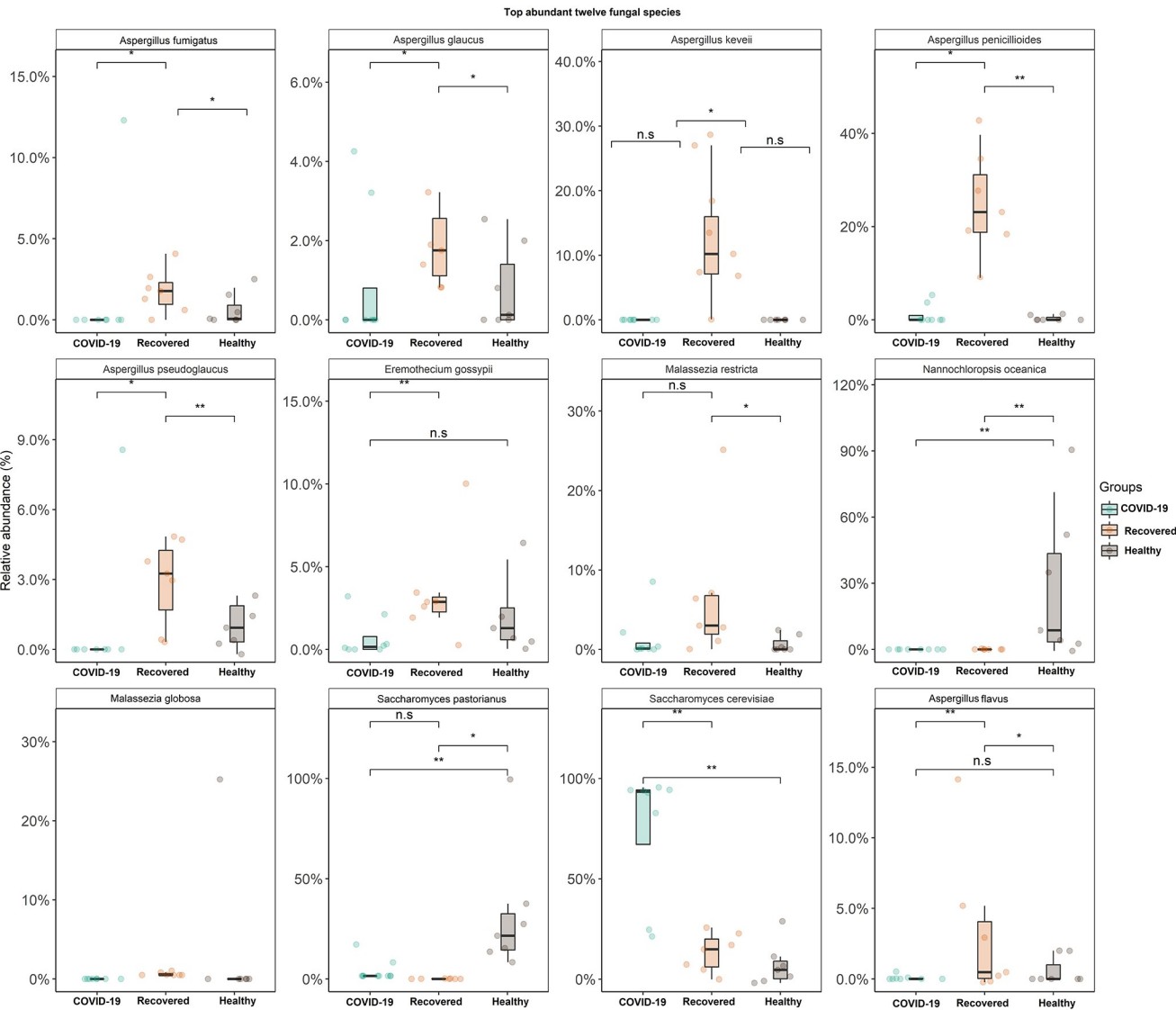

**Fig 6. Top twelve fungal species detected.** The species-level taxonomic abundance of mycobiome in COVID-19, Recovered and Healthy nasopharyngeal samples. The diversity for each species is plotted on boxplots and comparisons are made with pairwise Wilcoxon test rank sum tests. Significance level (p-value) 0.0001, 0.001, 0.01, 0.05, and 0.1 are represented by the symbols "****", "***", "**", "*", and "n.s", respectively.

## Differentially abundant fungal communities show positive correlation with metabolic functional perturbations

To shed light on whether SARS-CoV-2 infection could influence the metabolic functional potentials of the concurrent fungal mycobiota, we analysed the RNA-seq data through KEGG pathway and SEED subsystems. By examining the correlations between the different gene families (n = 33) of the same KEGG pathway for COVID-19 patients, Recovered humans, and Healthy control's NT mycobiome, we found significant differences (p = 0.01, Kruskal-Wallis test) in their composition and relative abundances (S2 Fig, S3 Table). Differentially abundant fungal species had significant correlations (i.e., positive or negative) with different KEGG pathways including cytokine-cytokine receptor functions, cellular processes, methane oxidation, malate dehydrogenase (mdh), oxidative phosphorylation, sulfur, protein and carbohydrates

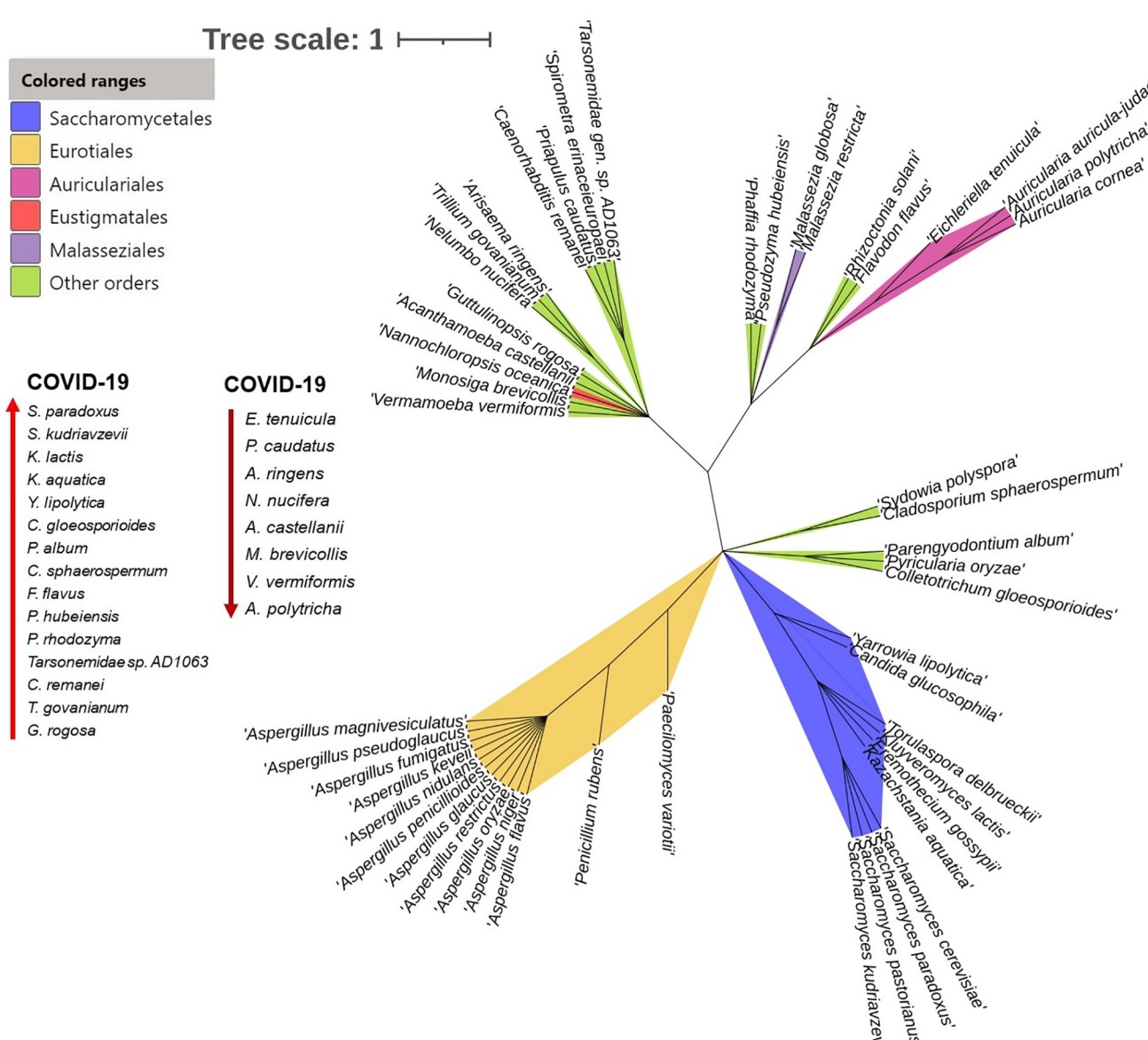

**Fig 7. Phylogenetic relationships among detected fungal communities.** Unrooted phylogenetic tree showing the relationship of 50 top abundant fungal species identified in Healthy people, COVID-19 patients and Recovered humans nasopharyngeal samples. Taxonomic groups indicated by the different color ranges show the phylogenetic origin and associations of different species in the respective fungal orders, with green for *Saccharomycetales*, yellow for *Eurotiales*, pink for *Auriculariales*, red for *Eustigmatales*, purple for *Malasseziales*, and green for other orders fungal orders. The upward arrow indicates the species those had opportunistic inclusion in COVID-19 patients and not merely detected in Healthy control samples. Conversely, the downward arrow denotes the Healthy people commensal species those are dysbiosed after SARS-CoV-2 infection (absent in COVID-19 patients). Bootstrap values are calculated based on 1000 replications and the tree scale in number of substitutions per site. The fungal species in the phylogenetic tree are also available in S1 Data.

metabolism and methionine degradation have positive correlation with the dominant fungal species (Fig 8A). For instance, cytokine-cytokine receptor functions revealed strongest positive correlation with *S. cerevisiae* (Spearman correlation; r > 0.65, p < 0.001); the top abundant fungal species in COVID-19 metagenome. Likewise, cellular processes showed significant positive associations with *A. pseudoglaucus* (Spearman correlation; r > 0.6, p < 0.01) and *M. globosa* (Spearman correlation; r > 0.5, p < 0.05); two dominating fungal species in Recovered

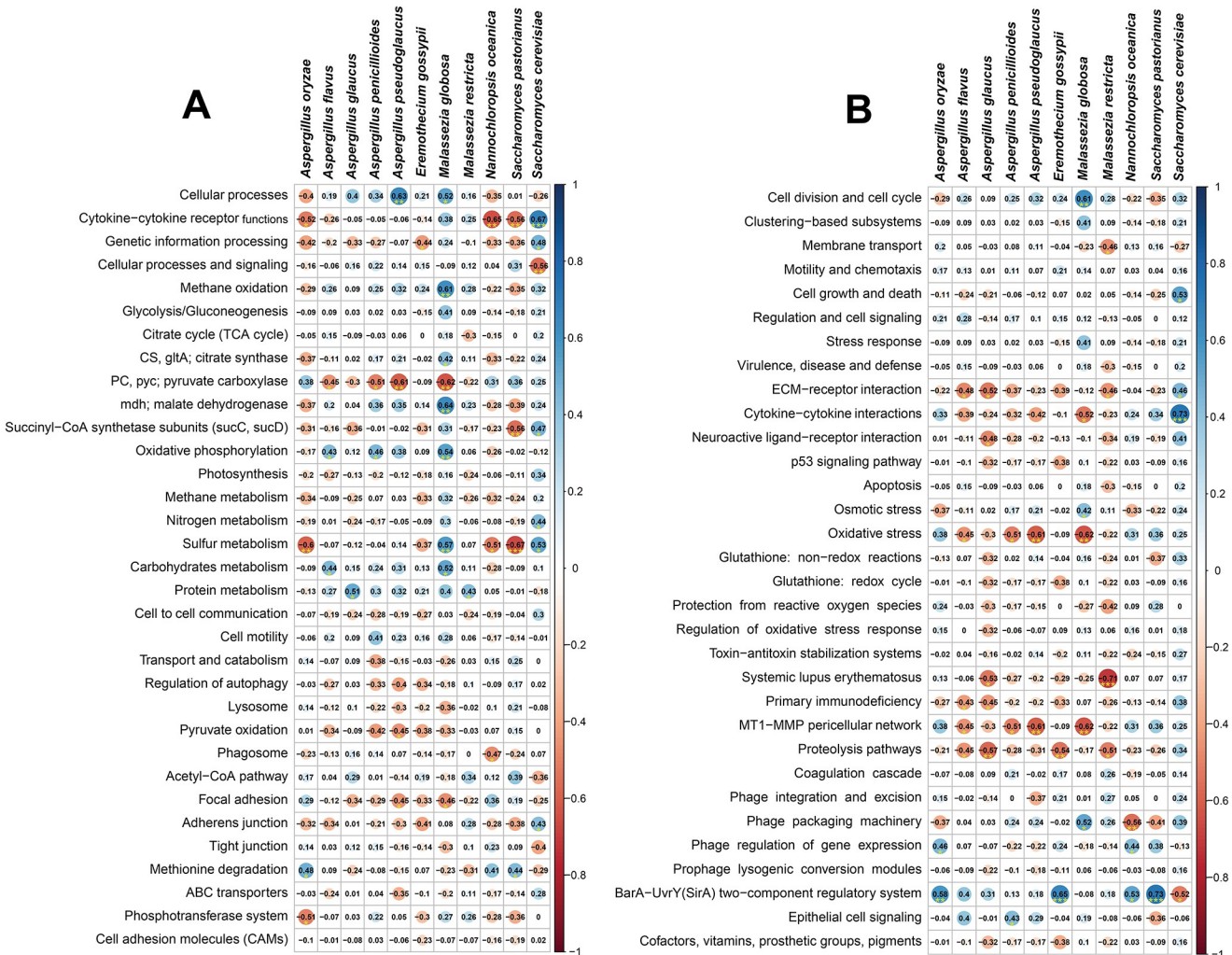

**Fig 8. Spearman's rank correlation matrix of the dominant fungal species and related metabolic functions.** (A) Correlation between KEGG orthologues (KOs) and predominant fungal species, and (B) correlation between SEED subsystems and top abundant fungal species. The numbers display pairwise Spearman's correlation coefficient (r). Blue and red colors indicate positive and negative correlation, respectively. The color density, circle size, and numbers reflect the scale of correlation. *Significant level (*p < 0.05; **p < 0.01; ***p < 0.001). The R packages, Hmisc (https://cran.r-project.org/web/packages/Hmisc/index.html) and corrplot (https://cran.r-project.org/web/packages/corrplot/vignettes/corrplot-intro.html) were used respectively to analyze and visualize the data.

samples. In addition, *M. globosa* displayed significant positive correlations (Spearman correlation; r > 0.4, p ≤ 0.05) with methane oxidation, mdh, oxidative phosphorylation and sulfur metabolism (Fig 8A). Conversely, two predominating fungal species in Healthy controls, *N. oceanica* (Spearman correlation; r > 0.5, p < 0.01) and *S. pastorianus* (Spearman correlation; r > 0.5, p < 0.01) had significant negative correlation with cytokine-cytokine receptor functions, sulfur metabolism, and succinyl-CoA synthetase subunits (sucC, sucD). Similarly, *A. oryzae*, one of the prevalent fungal species in Recovered humans had strong negative correlations (Spearman correlation; r ≥ 0.5, p ≤ 0.05) with cytokine-cytokine receptor functions, sulfur metabolism and phosphotransferase system. In addition, pyruvate carboxylase (pyc), phagosome activity and focal adhesion related metabolic activities also had significant negative correlations (Spearman correlation; r ≥ 0.4, p ≤ 0.05) with the top abundant fungal species of Recovered humans (Fig 8A).

We also sought to gain further insight into the SEED hierarchical protein functions, and found 37 statistically different (p = 0.013, Kruskal-Wallis test) subsystems in COVID (COVID-19 and Recovered) and Healthy control metagenomes. These SEED functions had significant correlations (either positive or negative) with the dominating fungal species in the respective metagenomes. As for example, the predominantly abundant fungal species in COVID-19 patients nasal swabs (*S. cerevisiae*) showed significant positive correlations with cytokine-cytokine interactions (Spearman correlation; r > 0.7, p <0.001), cell growth and death (Spearman correlation; r > 0.5, p <0.05), ECM-receptor interaction (Spearman correlation; r > 0.4, p <0.05). However, this species had also substantial negative association with BarA-UvrY(SirA) two-component regulatory system (Spearman correlation; r > 0.5, p <0.05) (Fig 8B). In contrast, BarA-UvrY(SirA) two-component regulatory system was found as the strongly correlated metabolic function in *S. pastorianus* (Spearman correlation; r > 0.7, p < 0.001), *E. gossypii* (Spearman correlation; r > 0.65, p < 0.001), *A. oryzae* (Spearman correlation; r > 0.55, p < 0.01) and *N. oceanica* (Spearman correlation; r > 0.5, p < 0.05). Likewise, *M. globosa* revealed significant positive correlation (Spearman correlation; r > 0.6, p < 0.01) with cell division and cell cycle related SEED functions (Fig 8B). Conversely, SEED functions including systemic lupus erythematosus, MT1-MMP pericellular network, proteolysis pathways, oxidative stress, ECM-receptor interaction and membrane transport had significant negative correlations (Spearman correlation; r ≥ 0.5, p < 0.05) with most of the top abundant fungal species in all three metagenomes (Fig 8B).

## Discussion

Emerging evidence indicated that SARS-CoV-2-infected individuals had an increased risk for coinfections. Therefore, the physicians need to be cognizant about excluding other treatable respiratory pathogens [11, 13, 41]. There are a great number of diverse beneficial commensal microorganisms constitutively colonizing the mucosal lining of the upper airway especially the nasopharyngeal tract (NT). These microbes comprise viruses, phages, bacteria, and fungi [11, 13, 42] that have elegant mutualistic relationships with the human host. In our previous study, we reported that SARS-CoV-2 infection induced dysbiosis of NT commensal bacteria, archaea and viruses with high inclusion of opportunistic pathobionts that elicite metabolic functional potential perturbations in COVID-19 patients [13]. In this study, we postulated that SARS-CoV-2 infection may also alter the NT commensal fungal population and diversity along with perturbations in their metabolic functions. To validate this hypothetical interplay between SARS-CoV-2 and resident commensal mycobiota in the nasal cavity of humans, we compared 22 high-throughput RNA-Seq data obtained from Healthy individuals, COVID-19 patients and Recovered humans.

The rapid development of automated, high-throughput sequencing methods including metagenomics, RNA-Seq, and bioinformatics [43] have made it possible to study the global biodiversity of fungi in various epidemiological niches including in COVI1D-9 patients. In the present study, we found a remarkable shift in the diversity and composition of the NT mycobiome in COVID-19 patients and Recovered humans compared to the Healthy controls. Although, several questions remain about defining the nature of dysbiosis for any particular fungal species, our present findings showed that SARS-CoV-2 infection reduces commensal fungal population with inclusion of pathobionts in the NT of human (Fig 3). Moreover, we detected a number of microbial genomic features, altered metabolic pathways, and functional genes associated with SARS-CoV-2 pathogenesis. Although it is well established that gut microbiota has a critical role in pulmonary immunity and host's defense against SARS-CoV-2 infection [44–46], this study for the first time determined the increased diversity and relative

abundances of mycobiota in the NT of humans due to the interactions of SARS-CoV-2 infection with resident mycobiome in human nasal cavity.

One of the most striking findings of the current study was the significant differences in mycobiome diversity between COVID-19 patients and Healthy controls nasal cavity keeping the closest relationship of fungal population in COVID-19 patients with Recovered humans. The COVID-19 patients NT mycobiome exhibited a statistically significant higher diversity (both within and between sample diversities) than those of Recovered humans and Healthy controls (Figs 1 and 2), which supports our hypothesis of dysbiosis and also agree with several recent reports [9, 47]. In contrast, most of the previous studies reported that SARS-CoV-2 infection reduces the bacterial diversity in COVID-19 patients respiratory tract and gut compared to their healthy counterpart [13, 48]. However, SARS-CoV-2 infection did not significantly alter mycobiome diversity in relation to the gender of the study population.

Results from the current analysis showed that *Ascomycota* was the most predominating fungal phylum in all of three metagenomes with highest relative abundances (~ 96.0%) in COVID-19 samples followed by Recovered humans (90.0%) and Healthy controls (88.0%). Our analysis revealed that COVID-19 patients exhibited a different composition of NT fungal communities than Recovered humans and Healthy controls. Six phyla, 52 orders, 213 genera and 315 species of fungi had sole association with the SARS-CoV-2 infection (Fig 3). We found vast differences in genus-level mycobiome signatures in nasal cavities of Healthy controls, COVID-19 patients and Recovered humans irrespective of the homogeneous genetic backgrounds and living status. For instance, *Nannochloropsis* had several-folds higher relative abundances in Healthy controls compared to COVID-19 and Recovered samples. Likewise, the COVID-19 patients and Recovered humans had several-folds higher abundance of *Saccharomyces* and *Aspergillus*, respectively than the Healthy controls. Moreover, the inter-individual variations in mycobiome signature (fungal genera) of participants was also observed. Remarkably, more than 43.0% fungal species had sole association with Healthy humans nasal microbiomes, which were not detected in COVID-19 patients and Recovered humans indicating the potential dysbiosis of these commensal species during the pathogenic magnitudes of SARS-CoV-2 infection. These results are corroborated with our previous study where we reported that 79% commensal bacterial species found in Healthy controls were not detected in COVID-19 and Recovered humans [13]. There are lines of emerging evidences that the mycobiome communities in different parts of the host body can be altered in relation to pathophysiological changes [31, 49, 50]. Our present findings are also consistent with several earlier studies which reported that interactions between SARS-CoV-2 and oral microbiomes [30], and SARS-CoV-2 and gut microbiomes [51] is associated with the pathophysiology of lung diseases.

Despite hundreds of thousands of fungal species, only a few causes disease in humans. In this study, *N. oceanica* and *S. pastorianus* were two predominantly abundant fungal species in Healthy human nasopharyngeal swabs. Different species of *Nannochloropsis* are eco-sustainable bioactive microalgae which can provide human with nutritional elements including polyunsaturated fatty acids (PUFAs), polyphenols, carotenoids and vitamins [52]. Recent evidences suggested that *N. oceanica* is a chief source of eicosapentaenoic acid (EPA, 20:5n-3) and docosahexaenoic acid (DHA, 22:6 n-3) recommended for humans use due to their beneficial effects including anti-atherogenic, anti-thrombotic and anti-inflammatory properties [52, 53]. On the other hand, *S. pastorianus* is a recently evolved interspecies hybrid of *Saccharomyces* genus commonly used in the brewing industry. Different species of *Saccharomyces* are recognised as beneficial probiotics that can help humans with IBS, Crohn's disease, diarrhoea, and a range of gastrointestinal infections [54, 55]. The co-evolution of humans and fungi suggests that complex mechanisms exist to allow the host immune system to respond to fungi [56].

One of the hallmark findings of the present study was the predominant association of *S. cerevisiae*, *P. rhodozyma* and *P. variotii* with SARS-CoV-2 infections in COVID-19 patients. A series of recent study suggested that *S. cerevisiae* should be considered as a potential opportunistic pathogen especially for patients with immunosuppression, cancer and other critical illnesses [54]. One of the latest studies demonstrated that the abundance *S. cerevisiae* in the guts of COVID-19 patients with fever were significantly higher than in COVID-19 patients with non-fever supporting our present results [57]. Bloodstream infection by *S. cerevisiae* in critically ill COVID-19 patients has recently been reported in several studies [55, 58]. *P. variotii* is considered the most prevalent agents of human infection, can affect various organ systems, primarily in immunocompromised patients or those with indwelling material [59]. Importantly, different species of *Aspergillus* were predominantly abundant in Recovered humans nasal cavity, and this shift in the fungal community is believed to be associated with re-establishment of eubiosis or a balanced NT microbiome after clearance of SARS-CoV-2. Although, *A. penicillioides* was detected as one of the top abundant fungal species in Recovered human nasal swab, this species has yet rarely been reported as a human pathogen except for cystic fibrosis in infants [60]. One of pioneering researches reported that *A. fumigatus* was the most common species causing coinfection in COVID-19 patients, followed by *A. flavus* and suggested that clinicians should keep alerting the possible occurrence of pulmonary aspergillosis in severe/critical COVID-19 patients [61, 62]. Saprophytic fungal species like *A. keveii* of the *Aspergillus* genus are the common contaminant of food and soil, their spores are ubiquitous and responsible for developing invasive aspergillosis in millions of humans each year [63]. *A. oryzae* is a low pathogenic fungus but may, like many other harmless microorganisms, grow in human tissue under exceptional circumstances [64]. In addition to *Aspergillus* spp., *S. cerevisiae* and *M. restricta* were also found to be dominating in the Recovered humans NT mycobiome. *Malassezia* spp. are lipid-dependent yeasts, inhabiting the skin and mucosa of humans and animals [65]. In adults, *M. restricta* and *M. globosa* are the major component of the healthy human skin mycobiome especially in Asia [66], and have not been linked to in the pathogenesis of any infectious diseases [67]. The interactions between human host and these fungal species can be mediated directly by specific pattern recognition receptors found on host cells and pathogen-associated molecular patterns present on fungal cell walls [66]. However, detailed clinical context is available for a very limited number of these species and data concerning their role in the pathophysiology of COVID-19 are even more scarce.

Remarkably, COVID-19 patients largely had inclusion of > 36.0% opportunistic fungal species, a part of commensal mycobiome that may become pathogenic in the event of host perturbation, such as dysbiosis or immunocompromised host. *P. rhodozyma*, one of the top abundant opportunistic fungal species found in COVID-19 patients nasal cavity, is an important microorganism for its use in both the pharmaceutical industries and food industry [68]. Although, numerous studies have addressed the molecular regulatory mechanisms of cell growth and astaxanthin synthesis by *P. rhodozyma* [69, 70], however, association of this species in disease causation has not been reported yet. Different species of *Saccharomycetales* such as *S. paradoxus*, *S. kudriavzevii*, *K. lactis*, *K. aquatica*, *Y. lipolytica* had an opportunistic inclusion in COVID-19 patients. Earlier evidences suggested that different species *Saccharomycetales* can only perform opportunistic or passive crossings when epithelial barrier integrity of the NT is previously compromised by other infectious agents [70]. For instance, *S. kudriavzevii* an opportunistic pathogen especially potential hazard to the health of immunocompromised workers in the wine industry, and potentially also to consumers [71]. *S. paradoxus*, mainly found in the wild environment, is the closest relative of the domesticated yeast *S. cerevisiae*. At least five different killer toxins are produced by *S. paradoxus* which can inhibit the growth of

other competing fungal species in immunocompromised host [72], and likely to cause opportunistic infection.

Despite the striking discrepancy in the phylogenetic composition and relative abundances of fungal species in three metagenomes, we found significant associations between differentially abundant fungal species and different metabolic functional pathways. Our findings revealed that enrichment of certain metabolic activities related to cytokine-cytokine receptor functions, cellular processes, methane oxidation, malate dehydrogenase, oxidative phosphorylation, sulfur, protein and carbohydrates metabolism, cell growth and death, and methionine degradation had strong positive correlation with 11 dominant fungal species, irrespective of the sample categories. The predominant fungal species of COVID-19 patients nasal cavity, *S. cerevisiae* was positively associated with cytokine-cytokine interactions, cell growth and death, and ECM-receptor interaction. These metabolic functional changes in COVID-19-associated mycobiome corroborated with previously reported other respiratory viral diseases [13, 73, 74]. Thus, our results provided evidence that enrichment of these metabolic activities are linked to consistent shifts in the structure and composition of the NT mycobiome with the progression of SARS-CoV-2 pathogenesis. Similar association was also found in the dominating fungi (e.g., *A. pseudoglaucus* and *M. globose*) of Recovered humans nasal cavity, in which metabolic functions like cellular processes, methane oxidation, malate dehydrogenase, oxidative phosphorylation and sulfur metabolism were found to be positively correlated. Based on this correlative evidence, it is tempting to speculate that SARS-CoV-2 infection associated shifts in mycobiome in the nasal cavity might also alter the metabolic functional potentials of the related microbiomes. Spearman's correlation analyses also revealed that the pyruvate carboxylase, phagosome activity, focal adhesion, MT1-MMP pericellular network, proteolysis pathways, oxidative stress, and membrane transport have significant negative correlations with the dominating fungal species. The metabolic health of an individual is represented by the proper functioning of organismal metabolic processes coordinated by multiple physiological systems [74]. The differentially abundant functions and pathways identified in this study corroborated with the findings from previous reports [75], and to COVID-19. However, some of the predicted metabolic features differed between COVID-19 and Healthy controls, perhaps representing metabolic changes associated with the progression of SARS-CoV-2 pathogenesis, and typical host-microbiome interactions in SARS-CoV-2 infected patients.

## Conclusions

Human nasopharyngeal microbiota plays a crucial role in providing protective responses against pathogens. Any alteration in the NT microbiota or their metabolites can cause immune dysregulation and impair the antiviral activity against respiratory viruses like SARS-CoV-2. Overall, the results of this study suggested that SARS-CoV-2 infection induces remarkable depletion of nasopharyngeal commensal fungal population with inclusion of different opportunistic pathogens in COVID-19 and Recovered samples. Several predicted functional pathways differed between COVID-19 patient and Recovered people nasopharyngeal samples compared to Healthy individuals which reflect the roles microbial metabolic changes with the progression of SARS-CoV-2 pathogenesis. These interactions were further complicated by the common co-existence of dominant mycobiota that interact with both host and SARS-CoV-2. The distinguishable fluctuations in the fungal population and associated genomic features detected in this study may serve as a benchmark for mycobiome-based diagnostic markers and formulating therapeutics for COVID-19 patients. Furthermore, it would be interesting to conduct future studies with a larger sample size to elucidate the modulation of commensal mycobiome, their functional potentials and genomic expression during the pathophysiology of

SARS-CoV-2 infection which will help in the development of specific therapeutic regimes against this pandemic disease.

## Supporting information

**S1 Fig. Major fungal phyla detected.** The phylum-level taxonomic profile of fungal microbiomes in COVID-19 (C1-C8), Recovered (R1-R7) and Healthy (H1-H7) nasopharyngeal samples. Phyla with $> 1\%$ mean relative abundance are represented by different color codes against respective sample groups. Others ($< 1\%$) indicates the rare taxa in each group, with mean relative abundance of $< 1\%$.
(PNG)

**S2 Fig. Metabolic functional potentials of the fungal microbiota.** Heatmap showing (A) KEGG orthologues (KOs) and (B) SEED subsystems associated with fungal metabolism in Healthy, COVID-19 and Recovered metagenomes.
(JPG)

**S1 Table. Demographic characteristics of the study people: Clinical diagnosis, treatment and recovery history of SARS-CoV-2 infections.**
(DOCX)

**S2 Table. Dysbiosis of mycobiomes after SARS-CoV-2 infections.**
(DOCX)

**S3 Table. Metabolic functional potentials of the fungal microbiome.**
(DOCX)

**S1 Data. Taxonomic information on fungal microbiomes.**
(XLSX)

## Acknowledgments

The authors would like to thank the Ministry of Science and Technology, Government of the People's Republic of Bangladesh for supporting this research. The authors would like to thank those who provided us the samples.

## Ethical approval

The protocol for sample collection from COVID-19, Recovered and Healthy humans, sample processing, transport, and RNA extraction was approved by the National Institute of Laboratory Medicine and Referral Center of Bangladesh. The study participants received a written informed consent letter consistent with the experiment.

## Author Contributions

**Conceptualization:** M. Nazmul Hoque, Md. Murshed Hasan Sarkar, M. Anwar Hossain, Tofazzal Islam.

**Data curation:** M. Nazmul Hoque, M. Shaminur Rahman, Shahina Akter, Tanjina Akhtar Banu, Barna Goswami.

**Formal analysis:** M. Nazmul Hoque, M. Shaminur Rahman.

**Funding acquisition:** Tanjina Akhtar Banu, Iffat Jahan, M. Salim Khan.

**Investigation:** Md. Murshed Hasan Sarkar, Md Ahashan Habib, Shahina Akter, Tanjina Akhtar Banu, Barna Goswami, Iffat Jahan, M. Salim Khan.

**Methodology:** M. Nazmul Hoque, M. Shaminur Rahman, Md. Murshed Hasan Sarkar, Md Ahashan Habib, Shahina Akter, Tanjina Akhtar Banu, Barna Goswami, Iffat Jahan, M. Salim Khan.

**Project administration:** Shahina Akter, Tanjina Akhtar Banu, Barna Goswami, Iffat Jahan, M. Salim Khan.

**Resources:** Md. Murshed Hasan Sarkar, Tanjina Akhtar Banu, Barna Goswami, Iffat Jahan, M. Salim Khan, Tofazzal Islam.

**Software:** M. Nazmul Hoque, M. Shaminur Rahman.

**Supervision:** M. Anwar Hossain, M. Salim Khan, Tofazzal Islam.

**Validation:** M. Nazmul Hoque, M. Shaminur Rahman, M. Anwar Hossain, Tofazzal Islam.

**Visualization:** M. Nazmul Hoque.

**Writing – original draft:** M. Nazmul Hoque.

**Writing – review & editing:** M. Anwar Hossain, Tofazzal Islam.

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
