## [Decision Letter · Decision Letter 0]

16 Aug 2022

PONE-D-22-14092Transcriptome analysis in nasopharyngeal samples reveals increased abundance and diversity of opportunistic fungal pathogens in COVID-19 patientsPLOS ONE

Dear Dr. Islam,

Thank you for submitting your manuscript to PLOS ONE. After careful consideration, we feel that it has merit but does not fully meet PLOS ONE’s publication criteria as it currently stands. Therefore, we invite you to submit a revised version of the manuscript that addresses the points raised during the review process.

We look forward to receiving your revised manuscript.

Kind regards,

David M. Ojcius

Academic Editor

PLOS ONE

Journal Requirements:

Reviewers' comments:

Reviewer's Responses to Questions

**Comments to the Author**

1. Is the manuscript technically sound, and do the data support the conclusions?

Reviewer #1: Yes

Reviewer #2: Yes

2. Has the statistical analysis been performed appropriately and rigorously? 

Reviewer #1: Yes

Reviewer #2: Yes

3. Have the authors made all data underlying the findings in their manuscript fully available?

Reviewer #1: Yes

Reviewer #2: Yes

4. Is the manuscript presented in an intelligible fashion and written in standard English?

Reviewer #1: Yes

Reviewer #2: Yes

5. Review Comments to the Author

Reviewer #1: This is a well-designed study with a good representation of data. However, there are some concerns and issues that need to be addressed by the authors before considering for publication in PLOS One.

Abstract

Rephrase the first sentence of the abstract and remove “we previously reported”.

Line 45: Surprising to see S. cerevisiae (88.62%) and Phaffia rhodozyma (10.30%) comprised 99% total species where authors reported 863 species.

Introduction:

1. Clinical trials and high throughput sequencing ………..respiratory viral pathogens [14], and bacteria and/or fungi [11, 13, 15, 16], make no sense. Rephrase the sentence.

2. Similarly, Fungal infections are known…….patients admitted to intensive care units with ARDS [11]. Revise the sentence to make it meaningful to the readers.

3. The hypothesis is time demanding and carries significant importance. However, the big sentence representing the hypothesis is very difficult to follow and coordinate. Please rephrase it to make a simpler and reader-friendly statement.

Method

1. The samples were collected more than two years ago. In the meantime, the variants of COVID have been changed a couple of times. In that case, what approach should be used by the authors to address concerns like currently circulating COVID strains?

Results

1. Our primary microbiome compositional analysis….. 11 species as differentially abundant across Healthy, COVID-19 and Recovered metagenomes. Make no sense. Rephrase it.

2. What is the necessity of mentioning major fungal species detected (Fig 5) and the top twelve fungal species detected (Fig 6) in two different figures? Isn’t it an exaggeration? Keep a single figure to represent major or top abundant fungal species and the rest may be replaced in the supplementary figures.

3. The Figures 5 and 6 legends for species names should be italic.

Discussion

1. Needs to correlate fungal infection with SARS-CoV2 infection in the discussion section.

2. Is there any report of secondary infection of fungus after COVID-19 infection?

3. Discuss fungal opportunistic pathogenic character in COVID-19 cases.

Reviewer #2: Summary

This study compared the nasopharyngeal fungal microbiome of 15 people: Eight COVID-19 patients, and seven healthy controls. Seven of the COVID-19 patients were subsequently included in the recovered group after tested negative and recovered from COVID-19.

The authors found that the three groups had unique fungal microbiome: 37% of fungal species were exclusively associated with SARS-CoV-2, with Saccharomyces cerevisiae and Phaffia rhodozyma being the two species with the highest abundance. The recovered patients’ fungal microbiome was dominated by several species of the Aspergillus genus, including A. penicillioides, A. keveii, A. oryzae, and A. pseudoglaucus.; Healthy controls had high abundance of Nannochloroopsis oceanica and Saccharomyces pastoriaus. Another main finding of the current study was that there was an increase in the alpha diversity of fungal microbiome in COVID-19 patients. The authors went on to perform metabolic functions analysis and postulated on the possible role of dysbiosis in the role of COVID-19 pathogenesis.

Overall Comment

This is an important study to allow deeper understanding of nasopharyngeal microbiome in COVID-19 patients. Extensive bioinformation analysis work was performed with interesting findings.

The description of patient recruitment could be made clearer. It took me some time to understand the 7 subjects in the recovered group were the sample subjects from the COVID-19 group, after they have recovered. The authors should also explain why only 7 of the 8 COVID-19 patients were included in the recovered group.

There was inconsistency in the terminology used for the three groups: For example: line 129 “Recovered”, line 133 “Recovered humans”, line 134 “Recovered subjects; Another example: line 138 “Healthy people”, line 136 “Healthy control subjects”.

The difference between dysbiosis and clinical infection was not clear throughout the manuscript. For example, in Lines 110-115, the authors stated the importance of understanding fungal microbiome in COVID-19 patients, however, in the next sentence, the authors advocated a timely diagnosis of fungal co-infections to limit the overuse of antimicrobial agents.

The authors included subject information in Table S1 with a column stated whether the subjects received “COVID-19 medicine”. Specific information on antibiotics used would be helpful to the interpretation of the study results: One major finding of the current study was that COVID-19 patients exhibited higher fungal microbiome diversities than those of recovered human and healthy controls, could it be due to effect of antibiotics, killing off most bacteria that allow the blooming of fungal organisms? Another major finding of the current study was that various species in the Aspergillus genus were over-represented in the recovered group. Aspergillus is well known to be present in the healthcare environment [1]. Could the observation be explained by nasopharynx flora being colonized by hospital molds with the aid of antibiotics therapy used?

Specific Comments

Lines 63 - 65: Redundance, can omit the second “SARS-CoV-2” of the sentence

Lines 65 – 68: The sentence seemed to suggest that all COVID-19 infections will result in ARDS. Suggest rephrasing.

Line 71: What is the meaning of “resilient microbiomes”?

Line 88: Mucormycosis is well studied and should not be considered as “mysterious fungal infection”

Lines 95-98: broad spectrum antibiotics are for treatment of bacterial coinfections, dexamethasone and immunosuppressive therapies are for immune modulation for treatment of immune dysregulation such as ARDS.

Lines 111: what is the meaning of “migration, propagation and immune response”?

Lines 250-251: “….which probably due to the opportunistic inclusion during the pathogenesis of SARS-CoV-2 infection” should be moved to discussion section

Line 314: Can consider delete the word “However”

Line 427: Typo: “COVI1D-9”

Line 488: redundance: “…..with SARS-CoV-2 infections in COVID-19 patients”

Line 502-505: Different species of aspergillus has different pathogenic potential. As the authors rightly pointed out A. fumigatus, and A. flavus to a lesser extent, are associated with clinical important aspergillus infections, but not the A. penicilliodes and A. keveii that is found in abundance in recovered patients. Suggest discussing pathogenic and non-pathogenic aspergillus species separately.

Line 570-572: was there any specific impairment of antiviral activities that was mentioned in the text?

Figures: some figures placed healthy control on the left side of the charts (e.g. Fig 1A, Fig 4), while others were in the opposite order, with COVID-19 on left side (e.g. Fig 3, Fig 5, and 6)

Figure 5: the color used for different species are similar to each other.

Table S1: Inconsistency noted: there were information included for 7 COVID-19 patients, 7 recovered patients, and 7 healthy controls. However, there were 8 COVID patients recruited in the study (line 37).

References

David J. Weber, MD, Amanda Peppercorn, Melissa B. Miller, Emily Sickbert-Benett, William A. Rutala, Preventing healthcare-associated Aspergillus infections: review of recent CDC/HICPAC recommendations, Medical Mycology, Volume 47, Issue Supplement_1, 2009, Pages S199–S209, https://doi.org/10.1080/13693780802709073

6. PLOS authors have the option to publish the peer review history of their article (what does this mean?). If published, this will include your full peer review and any attached files.

Reviewer #1: No

Reviewer #2: No

---

## [Author Response · Author response to Decision Letter 0]

15 Oct 2022

Point-by-point responses to the reviewer comments

Reviewer # 1

This is a well-designed study with a good representation of data. However, there are some concerns and issues that need to be addressed by the authors before considering for publication in PLOS One.

Abstract

Reviewer comment: Rephrase the first sentence of the abstract and remove “we previously reported”.

Our Response: We would like to thank the reviewer for complementing our manuscript, and suggestion for further revision. We are happy to revise and edit the Abstract as per reviewer’s suggestion. You may kindly go through the Abstract in the revised manuscript.

Reviewer comment: Line 45: Surprising to see S. cerevisiae (88.62%) and Phaffia rhodozyma (10.30%) comprised 99% total species where authors reported 863 species. 

Our Response: Thank you very much for this question. We detected 533 fungal species in COVID-19 patients NT samples (out of 863 species detected in all of three groups). Among these species S. cerevisiae (88.62%) and Phaffia rhodozyma (10.30%) were the two top abundant species contributing about ~99.0% of total species, and rest of the species had <0.5% relative abundances. Higher abundance of Saccharomyces organisms in COVID-19 patients could be due to invasive infection in immunocompromised or critically ill patients. Severe bloodstream infection by Saccharomyces in hospitalized ICU patients, due to severe COVID-19, has been reported previously (Ventoulis et al., 2020; DOI: 10.3390/jof6030098). Furthermore, oral administration of immunomodulatory component derived from S. cerevisiae reduced intestinal inflammation and promoted the reduction of overgrowth of other fungal species such as Candida glabrata, C. albicans etc. in the gut (Jawhara et al., 2020; DOI: 10.1186/s13099-020-00385-2). One of the latest studies demonstrated that the abundance S. cerevisiae in the guts of COVID-19 patients with fever were significantly higher than in COVID-19 patients with non-fever supporting our present results (Zhou et al., 2021; DOI: 10.2147/JIR.S311518). We have well discussed this issue in the Discussion section. You may kindly go through Lines 518-528 in the Discussion section of the revised manuscript.

Introduction

Reviewer comment: Clinical trials and high throughput sequencing ………..respiratory viral pathogens [14], and bacteria and/or fungi [11, 13, 15, 16], make no sense. Rephrase the sentence.

Our Response: We would like to thank the reviewer for this nice suggestion. We have rephrased the sentence You may kindly go through Lines 77-79 in the revised manuscript.

Reviewer comment: Similarly, Fungal infections are known…….patients admitted to intensive care units with ARDS [11]. Revise the sentence to make it meaningful to the readers.

Our Response: We would like to thank the reviewer for this nice suggestion. We have rephrased the sentence You may kindly go through Lines 85-87 in the revised manuscript.

Reviewer comment: The hypothesis is time demanding and carries significant importance. However, the big sentence representing the hypothesis is very difficult to follow and coordinate. Please rephrase it to make a simpler and reader-friendly statement.

Our Response: We would like to thank the reviewer for this valid suggestion. We have revised the hypothesis to making a simpler and reader-friendly statement. You may kindly go through Lines 113-116 in the revised manuscript.

Method

Reviewer comment: The samples were collected more than two years ago. In the meantime, the variants of COVID have been changed a couple of times. In that case, what approach should be used by the authors to address concerns like currently circulating COVID strains?

Our Response: We would like to thank the reviewer for this critical concern. We do agree with the Reviewer’s opinion. Unfortunately, we are too late (due to some unavoidable issues) to publish this article timely. It would be good if we could publish the findings earlier. SARS-CoV-2 is the fast-evolving virus, and altering genetic mutations of SARS-CoV-2 variants has decreased the effectiveness of therapeutics and vaccines. SARS-CoV-2 can mutate in individuals, and these variants can be propagated across populations over time. Although, scattered information on fungal co-infections in SARS-CoV-2 is available, however, no study has elaborated on the association between gut microbiota and fungal microbiomes in COVID-19 patients. Therefore, despite the emergence of different variants of SARS-COV-2 after we collected sample, our data/findings of the present study would increase our understanding on early interaction of this virus (SARS-CoV-2) with co-infecting microbes (e.g., fungi) of the host. The present findings could shed light on developing microbiome-based diagnostics, and also devising appropriate therapeutic regimens including antifungal drugs for prevention and control of concurrent fungal coinfections in COVID-19 patients. We hope that the judicious Reviewer will be considerate to accept our limitation in this regard.

Results 

Reviewer comment: Our primary microbiome compositional analysis. 11 species as differentially abundant across Healthy, COVID-19 and Recovered metagenomes. Make no sense. Rephrase it.

Our Response: We would like to thank the reviewer for this valid suggestion. We have rephrased the sentence to making it a reader-friendly statement. You may kindly go through Lines 341-342 in the revised manuscript.

Reviewer comment: What is the necessity of mentioning major fungal species detected (Fig 5) and the top twelve fungal species detected (Fig 6) in two different figures? Isn’t it an exaggeration? Keep a single figure to represent major or top abundant fungal species and the rest may be replaced in the supplementary figures.

Our Response: We would like to thank the reviewer for this concern. Figure 5 represents the species-level taxonomic profile (top abundant 11 species) of fungal microbiomes in COVID-19 (C1-C8), Recovered (R1-R7) and Healthy (H1-H7) nasopharyngeal samples. Species with > 1% relative abundance are represented against respective sample groups, and rest of the species are indicated as less abundant taxa in each group (i.e., Other species; < 1%). Conversely, Figure 6 shows the pairwise statistical relationship of 12 differentially fungal species and health biomarkers (COVID-19 patients, Recovered humans and Healthy controls) of the study participants with significant variations (p ≤ 0.05, Wilcoxon test). Both number and name of fungal species differ in both figures. Therefore, we firmly believe that both figures carry significant information for the readers. We hope that the sensible Reviewer will agree with us to keep both Figure 5 and Figures 6 as main figures, independently. 

Reviewer comment: The Figures 5 and 6 legends for species names should be italic.

Our Response: We would like to thank the reviewer for this valuable suggestion. We have revised both figures accordingly. You may kindly see Figure 5 and Figure 6 in the revised manuscript.

Discussion

Reviewer comment: Needs to correlate fungal infection with SARS-CoV2 infection in the discussion section.

Our Response: We would like to thank the reviewer for this valuable suggestion. We have revised and edited the discussion section with correlation between SARS-CoV2 infection and fungal co-infection. You may kindly go through Lines 447-460 in the revised manuscript.

Reviewer comment: Is there any report of secondary infection of fungus after COVID-19 infection?

Our Response: Thank you very much for this important question. Yes, there are reports of secondary infection of fungus after SARS-CoV-2 infection. For instance, severe bloodstream infection by Saccharomyces in hospitalized ICU patients, due to severe COVID-19, has been reported previously (Ventoulis et al., 2020; DOI: 10.3390/jof6030098). Moreover, aspergillosis, invasive candidiasis, and mucormycosis (black fungus) have been reported as the most commonly reported fungal infections in patients with COVID-19 (Hoenigl M, 2021; DOI: 10.1093/cid/ciaa1342, Gangneux et al., 2020; DOI: 10.1016/j.mycmed.2020.100971).

Reviewer comment: Discuss fungal opportunistic pathogenic character in COVID-19 cases.

Our Response: We would like to thank the reviewer for this nice comment. We have discussed the role of opportunistic fungal infections amid COVID-19 in several paragraphs of the discussion section. You may kindly go through Lines 447-460 and 518-528 in the revised manuscript.

Reviewer ## 2

Overall Comment

This is an important study to allow deeper understanding of nasopharyngeal microbiome in COVID-19 patients. Extensive bioinformation analysis work was performed with interesting findings.

Our Response: We would like to express our sincere thanks to the expert reviewer for complementing our manuscript.

Reviewer comment: The description of patient recruitment could be made clearer. It took me some time to understand the 7 subjects in the recovered group were the sample subjects from the COVID-19 group, after they have recovered. The authors should also explain why only 7 of the 8 COVID-19 patients were included in the recovered group.

Our Response: We would like to thank the reviewer for this valid suggestion. We have revised the methodology of subject recruitment. The confirmed COVID-19 patients (n = 8) were admitted into the dedicated COVID-19 isolation wards, and received medication. Unfortunately, one confirmed COVID-19 patient died in the ICU (intensive care unit) during the course of medication (six days after confirmatory diagnosis). These rest of the patients (n = 7) were tested negative for COVID-19 after 17.5 (ranged from11 to 32) days of SARS-CoV-2 infection, and categorized as Recovered humans (S1 Table). You may kindly go through Lines 142-148 in the revised manuscript.

Reviewer comment: There was inconsistency in the terminology used for the three groups: For example: line 129 “Recovered”, line 133 “Recovered humans”, line 134 “Recovered subjects; Another example: line 138 “Healthy people”, line 136 “Healthy control subjects”.

Our Response: We would like to thank the reviewer for pointing out this inconsistency. We revised the manuscript using unique terminology for three groups i.e., COVID-19 patients, Recovered humans and Healthy controls. We hope that the judicious reviewer will find no more inconsistency regarding group names throughout the manuscript. You may kindly go through the revised manuscript

Reviewer comment: The difference between dysbiosis and clinical infection was not clear throughout the manuscript. For example, in Lines 110-115, the authors stated the importance of understanding fungal microbiome in COVID-19 patients, however, in the next sentence, the authors advocated a timely diagnosis of fungal co-infections to limit the overuse of antimicrobial agents.

Our Response: We would like to thank the reviewer for this valid comment. We have deleted the conflicting sentence from the manuscript, and added few statements on fungal dysbiosis and clinical infection of SARS-CoV-2. You may kindly go through Lines 97-101 in the revised manuscript.

Reviewer comment: The authors included subject information in Table S1 with a column stated whether the subjects received “COVID-19 medicine”. Specific information on antibiotics used would be helpful to the interpretation of the study results: One major finding of the current study was that COVID-19 patients exhibited higher fungal microbiome diversities than those of recovered human and healthy controls, could it be due to effect of antibiotics, killing off most bacteria that allow the blooming of fungal organisms? Another major finding of the current study was that various species in the Aspergillus genus were over-represented in the recovered group. Aspergillus is well known to be present in the healthcare environment [1]. Could the observation be explained by nasopharynx flora being colonized by hospital molds with the aid of antibiotics therapy used?

Our Response: We would like to thank the reviewer for this valuable information and suggestion. We have added the suggested information (i.e., medication) in the method, results and discussion sections of the manuscript. The confirmed COVID-19 patients received medication with broad spectrum antibiotics (e.g., azithromycin and cefuroxime) and ivermectin for an average period of 15.71 days. One of the major findings of the current study was that COVID-19 patients exhibited higher fungal microbiome diversities than those of Recovered humans and Healthy controls, which could be due to the effect of medication with broad-spectrum antibiotics (e.g., azithromycin and cefuroxime), and thereby, killing off most bacteria that allow the blooming of fungal organisms. Another important finding of the present study was the detection of various species in the Aspergillus genus with higher relative abundances in the Recovered humans group. Aspergillus is well known to be present in the healthcare environment [44], and thus, could colonize in the nasopharyngeal cavity of the patients by invasive molds commonly found within a healthcare facility. You may kindly go through Lines 97-121, 230-232, and 447-460 in the revised manuscript.

Specific comments

Lines 63 - 65: Redundance, can omit the second “SARS-CoV-2” of the sentence

Our Response: We would like to thank the reviewer for this valid suggestion. We have deleted the sentence/lines. You may kindly go through the revised manuscript.

Lines 65 – 68: The sentence seemed to suggest that all COVID-19 infections will result in ARDS. Suggest rephrasing.

Our Response: We would like to thank the reviewer for this important comment. We have rephrased the mentioned lines. You may kindly go through Lines 64-71 in the revised manuscript.

Line 71: What is the meaning of “resilient microbiomes”?

Our Response: Thank you for this nice question. Resilient microbiomes mean healthy commensal microbiota. However, we have replaced the words with healthy commensal microbiomes. You may kindly go through Line 73 in the revised manuscript.

Line 88: Mucormycosis is well studied and should not be considered as “mysterious fungal infection”

Our Response: We would like to thank the reviewer for this valid suggestion. We have rephrased the Line. You may kindly go through Line 89-90 in the revised manuscript.

Lines 95-98: broad spectrum antibiotics are for treatment of bacterial coinfections, dexamethasone and immunosuppressive therapies are for immune modulation for treatment of immune dysregulation such as ARDS.

Our Response: We would like to thank the reviewer for this concern. We have paraphrased the mentioned lines. You may kindly go through Lines 97-101 in the revised manuscript.

Lines 111: what is the meaning of “migration, propagation and immune response”?

Our Response: We would like to thank the reviewer for this valid suggestion. We have revised the Line. You may kindly go through Lines 113-116 in the revised manuscript.

Lines 250-251: “….which probably due to the opportunistic inclusion during the pathogenesis of SARS-CoV-2 infection” should be moved to discussion section

Our Response: We would like to thank the reviewer for this valid suggestion. We have shifted this statement in the Discussion section. You may kindly go through Lines 261-264 and 447-460 in the revised manuscript.

Line 314: Can consider delete the word “However”

Our Response: We would like to thank the reviewer for this valid suggestion. The word has been deleted. You may kindly see Line 330 in the revised manuscript.

Line 427: Typo: “COVI1D-9”

Our Response: Thank you very much. We’ve corrected the typo-mistakes in the mentioned line. You may kindly go through Line 444 in the revised manuscript.

Line 488: redundance: “…..with SARS-CoV-2 infections in COVID-19 patients”

Our Response: We would like to thank the reviewer for this noticing this redundancy. We have deleted the redundant words from the mentioned lines. You may kindly go through Lines 518-519 in the revised manuscript.

Line 502-505: Different species of aspergillus has different pathogenic potential. As the authors rightly pointed out A. fumigatus, and A. flavus to a lesser extent, are associated with clinical important aspergillus infections, but not the A. penicilliodes and A. keveii that is found in abundance in recovered patients. Suggest discussing pathogenic and non-pathogenic aspergillus species separately.

Our Response: We would like to thank the reviewer for this nice suggestion. We have discussed the issue of pathogenic and non-pathogenic aspergillus species separately. You may kindly go through Lines 518-528 in the revised manuscript. 

Line 570-572: was there any specific impairment of antiviral activities that was mentioned in the text?

Our Response: We would like to thank the reviewer for this nice query. We have deleted the confusing statement. You may kindly go through the revised manuscript.

Figures: some figures placed healthy control on the left side of the charts (e.g. Fig 1A, 1B), while others were in the opposite order, with COVID-19 on left side (e.g. Fig 3, Fig 5, and 6)

Our Response: We would like to thank the reviewer for this inconsistency. We are happy to revise the mentioned Figures keeping uniformity. You may kindly see the revised Figures in the revised manuscript.

Figure 5: the color used for different species are similar to each other.

Our Response: Thank you for this concern. We have revised the Figure. You may kindly see the Figure 5 in the revised manuscript.

Table S1: Inconsistency noted: there were information included for 7 COVID-19 patients, 7 recovered patients, and 7 healthy controls. However, there were 8 COVID patients recruited in the study (line 37).

Our Response: We would like to thank the reviewer for noticing inconsistency. We have revised the Table S1 including the information of COVID-8 (ICU death). You may kindly see Table S1, and the revised manuscript.

---

## [Decision Letter · Decision Letter 1]

10 Nov 2022

Transcriptome analysis in nasopharyngeal samples reveals increased abundance and diversity of opportunistic fungal pathogens in COVID-19 patients

PONE-D-22-14092R1

Dear Dr. Islam,

We’re pleased to inform you that your manuscript has been judged scientifically suitable for publication and will be formally accepted for publication once it meets all outstanding technical requirements.

Kind regards,

David M. Ojcius

Academic Editor

PLOS ONE

Additional Editor Comments (optional):

Reviewers' comments:

Reviewer's Responses to Questions

**Comments to the Author**

1. If the authors have adequately addressed your comments raised in a previous round of review and you feel that this manuscript is now acceptable for publication, you may indicate that here to bypass the “Comments to the Author” section, enter your conflict of interest statement in the “Confidential to Editor” section, and submit your "Accept" recommendation.

Reviewer #1: All comments have been addressed

Reviewer #2: All comments have been addressed

2. Is the manuscript technically sound, and do the data support the conclusions?

Reviewer #1: Yes

Reviewer #2: Yes

3. Has the statistical analysis been performed appropriately and rigorously? 

Reviewer #1: Yes

Reviewer #2: Yes

4. Have the authors made all data underlying the findings in their manuscript fully available?

Reviewer #1: Yes

Reviewer #2: Yes

5. Is the manuscript presented in an intelligible fashion and written in standard English?

Reviewer #1: Yes

Reviewer #2: Yes

6. Review Comments to the Author

Reviewer #1: Authors addressed my comments and concerns with sufficient details. Though there are some limitations including sampling time (two-years back), however the findings are interesting and significant in terms of fungal pathogens in COVID19 infection.

Reviewer #2: The authors had adequately addressed all the comments and concerns I raised previously. The revised manuscript has improved significantly and is good for publication.

7. PLOS authors have the option to publish the peer review history of their article (what does this mean?). If published, this will include your full peer review and any attached files.

Reviewer #1: No

Reviewer #2: No

---

## [Editor Report · Acceptance letter]

5 Jan 2023

PONE-D-22-14092R1 

Transcriptome analysis reveals increased abundance and diversity of opportunistic fungal pathogens in nasopharyngeal tract of COVID-19 patients 

Dear Dr. Islam:

I'm pleased to inform you that your manuscript has been deemed suitable for publication in PLOS ONE. Congratulations! Your manuscript is now with our production department. 

Kind regards, 

on behalf of

Dr. David M. Ojcius 

Academic Editor

PLOS ONE